# Continual Low-Rank Adapters for LLM-based Generative Recommender Systems

**Hyunsik Yoo**[†]**, Ting-Wei Li**[†]**, SeongKu Kang**[‡]**, Zhining Liu**[†]**,**
**Charlie Xu**[§]**, Qilin Qi**[§]**, Hanghang Tong**[†]
[†]University of Illinois Urbana-Champaign    [‡]Korea University    [§]Amazon
`{hy40, twli, liu326, htong}@illinois.edu`
`seongkukang@korea.ac.kr  {caizhx, qilinqi}@amazon.com`

## Abstract

While large language models (LLMs) achieve strong performance in recommendation, they face challenges in continual learning as users, items, and user preferences evolve over time. Existing LoRA-based continual methods primarily focus on preserving performance on previous tasks, but this overlooks the unique nature of recommendation: the goal is not to predict past preferences, and outdated preferences can even harm performance when current interests shift significantly. To address this, we propose PESO (Proximally rEgularized Single evolving lOra), a continual adaptation method for LoRA in recommendation. PESO introduces a proximal regularizer that anchors the current adapter to its most recent frozen state, enabling the model to flexibly balance adaptation and preservation, and to better capture recent user behaviors. Theoretically, we show that this proximal design provides data-aware, direction-wise guidance in the LoRA subspace. Empirically, PESO consistently outperforms existing LoRA-based continual learning methods. Our code is available at https://github.com/hsyoo32/peso.

## 1 Introduction

Large language models (LLMs) are increasingly used for recommendation by treating the task as sequence generation: given a user's interaction history, the model autoregressively generates the next item tokens (Bao et al., 2025a; Cao et al., 2024; Tan et al., 2024; Wang et al., 2024; Bao et al., 2023; Kweon et al., 2025; Lin et al., 2025; 2026; Wei et al., 2025a). In practice, LLM is fine-tuned on user histories paired with their next interactions, aligning it with the recommendation objective. However, real-world interaction data are continuously collected and evolve over time: new users and items appear, and user preferences drift (Li et al., 2025a; Qiu et al., 2024). Periodic retraining from scratch on both historical and new data is possible but highly inefficient, making *continual learning* (i.e., updating the model effectively with new data) a natural and appealing solution.

It is well known that a continual model must balance *stability* (retaining past knowledge) and *plasticity* (adapting to new knowledge) (Zhu et al., 2021; Arani et al., 2022; Ye et al., 2022; Zhang et al., 2024a; Yuan et al., 2021; Do & Lauw, 2023; Mi et al., 2020; Yoo et al., 2025b). However, continual recommender systems present unique interpretations of these concepts, and bear subtle but critical difference from other domains such as computer vision (Bao et al., 2025b; Zeng et al., 2026a). In most other domains, continual tasks are typically disjoint and not time-ordered (e.g., cats vs. dogs → trucks vs. sedans), and the primary objective is to preserve performance on previous tasks (stability) while adapting to new ones (plasticity). In contrast, the ultimate goal of continual recommendation is to accurately capture evolving user preferences in order to predict which items a user *will* prefer in the near future. That is, recommendation is not concerned with predicting past user preferences; in fact, outdated preferences can even hinder performance if current user interests have shifted significantly (e.g., a user starts preferring romance over action). Thus, stability in recommendation refers to preserving long-term user preferences (e.g., enduring interests in certain genres or brands) that remain predictive, even if they are not strongly reflected in recent data. Plasticity, on the other hand, is required to overwrite outdated preferences and to capture emerging trends. This distinct setting in turn requires careful model design.

A common recipe for fine-tuning LLMs in recommendation is Low-Rank Adaptation (LoRA) (Hu et al., 2022; Liu et al., 2025a; Zeng et al., 2025; Qiu et al., 2026), due to simplicity and modularity across components (e.g., attention layers). LoRA freezes pretrained weights and injects lightweight trainable low-rank matrices. Its efficiency makes LoRA a natural candidate for continual learning, motivating our focus on continual LoRA for LLM-based recommenders. A simple intuitive approach is to maintain a *single evolving LoRA*: sequentially fine-tuning one adapter, initializing it from the previous stage and optimizing it on new data. This provides strong plasticity while parameter inheritance provides partial preservation of past knowledge. However, it inevitably overwrites useful past knowledge during fine-tuning, leading to forgetting.

To mitigate forgetting, several works in vision have proposed the family of *cumulative LoRA* (Wu et al., 2025; Liang & Li, 2024; Lu et al., 2024; Wang et al., 2023a; Liu et al., 2024a), which typically use the sum of the new trainable adapter and all frozen past adapters. This design explicitly enhances stability by reusing prior adapters and expanding LoRA's effective capacity, and it works well when tasks are largely independent (i.e., with minimal interference), allowing each adapter to encode task-specific knowledge. Intuitively, this might seem beneficial for recommendation, where preserving useful past preferences matters. However, our analysis shows that cumulative LoRA often underperforms the simpler single evolving LoRA. Unlike vision tasks, recommendation involves reappearing users with continuously evolving preferences. The model must therefore capture useful interference across stages, but frozen adapters entangle outdated and relevant preferences, making them hard to disentangle. In addition, as adapters accumulate over time, cumulative LoRA incurs growing storage costs and struggles to reflect their relative importance during aggregation.

To address these limitations, we adopt two principles: (1) avoid multiple adapters, which implicitly assume task independence, and (2) preserve past knowledge in a way that supports understanding of current user behavior. Guided by this, we propose PESO (Proximally rEgularized Single evolving lOra), which maintains a single evolving LoRA adapter while regularizing it toward its past state with a lightweight proximal term. Unlike cumulative LoRA, PESO balances stability and plasticity through the natural competition between the data-fitting loss and the proximal term, allowing the model to decide what to adapt or retain. Theoretically, we show that this design yields data-aware, direction-wise guidance in the LoRA subspace. We further instantiate it with a permodule softmax–Kullback–Leibler (KL) proximal, which preserves internal module structure rather than treating all parameters equally (i.e., a more nuanced stability mechanism). Empirically, PESO consistently outperforms both cumulative LoRA and the single evolving adapter across multiple real-world datasets, achieving a more effective stability–plasticity balance for recommendation.

In summary, our main contributions are threefold. **(1) Analysis:** we identify the distinctive stability–plasticity challenge in continual recommendation and show empirically that cumulative LoRA, while effective in simulated user-disjoint settings, underperforms in the natural case where user preferences evolve across time stages; **(2) Method and Theory:** we propose PESO, a *proximally regularized LoRA* that anchors each update to the previous state, with theory showing direction-wise, data-aware guidance and a per-module softmax–KL instantiation; **(3) Experiments:** we demonstrate through extensive experiments on real-world datasets that PESO consistently outperforms both single evolving and cumulative LoRA.

## 2 PRELIMINARY

**Notations.** We consider an LLM-based recommender that, given a user's interaction history, autoregressively predicts the next item token. At time stage $t \in \{1, \ldots, T\}$, let $\mathcal{U}_t$ be the set of active users, $\mathcal{I}_t$ the item set, and let $\mathcal{E}_t = \{(x_{u,1}, \ldots, x_{u,N_u})\}_{u \in \mathcal{U}_t}$ denote the collection of user interaction sequences, where $x_{u,n} \in \mathcal{I}_t$. For notational simplicity, we present stage-$t$ data as one prefix-target pair per user:

$$\mathcal{D}_t = \{(\mathbf{x}_u, y_u) : u \in \mathcal{U}_t, \ N_u \geq 2\}, \quad \mathbf{x}_u = (x_{u,1}, \ldots, x_{u,N_u-1}), \quad y_u = x_{u,N_u}. \tag{1}$$

In practice, training often uses multiple sliding-window next-item pairs induced from each sequence (see Appendix C.1). Each item is represented by *semantic ID* obtained by a codebook-based tokenizer (e.g., RQ-VAE, Rajput et al., 2023) trained on item semantic features (e.g., title/description), yielding fixed number of token IDs for each item. Semantic ID captures hierarchical semantics of items and works well in practice.[1]

---

[1]Adapting the tokenizer to new items over time is an interesting direction; here we fix the item tokenizer to isolate continual adaptation of the model (LoRA).

**Stability and Plasticity in Continual Recommendation.** We assume an initial model is pretrained offline on base data $\mathcal{D}_1$, and then fine-tuned sequentially on chronologically arriving blocks $\mathcal{D}_2, \ldots, \mathcal{D}_T$. The goal of continual recommendation is to minimize predictive risk on future interactions by balancing *stability* (retaining persistent long-term preferences) and *plasticity* (adapting to new or shifting preferences from recent data), thereby capturing evolving user interests (see Appendix A for a formal conceptual model). Concretely, for $\mathcal{D}_t$, the LLM is fine-tuned with the standard autoregressive cross-entropy for next-item prediction. Let $y = (y_1, \ldots, y_M)$ denote the semantic-ID token sequence of the target item. The stage-$t$ training loss is

$$L_{\text{ce}}^{\mathcal{D}_t} = \mathbb{E}_{(x,y) \sim \mathcal{D}_t} \big[ -\log p_\theta(y \mid x) \big], \qquad p_\theta(y \mid x) = \prod_{m=1}^{M} p_\theta(y_m \mid x, y_{<m}). \qquad (2)$$

**Low-Rank Adaptation (LoRA).** LoRA freezes the pretrained LLM weight $W_0 \in \mathbb{R}^{d_{\text{out}} \times d_{\text{in}}}$ and adds a trainable low-rank update:
$$\Delta W = BA, \quad A \in \mathbb{R}^{r \times d_{\text{in}}}, \ B \in \mathbb{R}^{d_{\text{out}} \times r}, \ r \ll \min(d_{\text{in}}, d_{\text{out}}), \qquad (3)$$

so that for an input $x \in \mathbb{R}^{d_{\text{in}}}$ the layer computes $(W_0 + \Delta W)\,x$. Only $A$ and $B$ are updated during fine-tuning, while $W_0$ remains fixed. This yields substantial parameter savings and modular, layer-wise adaptation (e.g., on attention projections). In this work, our analysis and method operate entirely within this LoRA subspace and therefore inherit its efficiency. We now formally define our problem.

**Problem 1.** *(Continual adaptation of a generative recommender)* ***Given:*** *(1) a pretrained LLM-based recommendation model (fine-tuned with LoRA on $\mathcal{D}_1$), (2) a sequence of chronological data blocks $\mathcal{D}_2, \ldots, \mathcal{D}_T$;* ***Goal:*** *learn updates that, at each stage $t$, adapt the model to $\mathcal{D}_t$ while retaining useful knowledge from earlier stages, achieving high quality next-item recommendation via a balanced stability–plasticity.*

## 3 ANALYSIS OF LoRA VARIANTS FOR CONTINUAL RECOMMENDATION

We introduce two primary baselines for our problem: single evolving LoRA and the cumulative LoRA family. Then, we empirically compare them on a natural chronological split and a user-disjoint split.

**Single evolving LoRA.** At stage $t$, the LoRA matrices $A_t$ and $B_t$ are initialized (i.e., parameter inheritance) from the previous stage ($A_{t-1}$ and $B_{t-1}$) and fine-tuned on new data $\mathcal{D}_t$:

$$W_t = W_0 + B_t A_t, \qquad B_t \leftarrow B_{t-1}, A_t \leftarrow A_{t-1}, \qquad t \geq 2, \qquad (4)$$

where $W_0$ is the pretrained LLM weight (i.e., not LoRA updates). This baseline is simple and adapts effectively to new data, while parameter inheritance provides partial preservation of past knowledge at initialization. However, it inevitably overwrites useful past knowledge during fine-tuning, leading to forgetting.

**Cumulative LoRA Variants**. To mitigate forgetting, cumulative LoRA has been widely used in domains such as vision (Wu et al., 2025; Liang & Li, 2024). At stage $t$, it reuses frozen adapters from past stages and adds a new trainable adapter by summing them during both training and inference. The effective update is

$$W_t = W_0 + \sum_{i=1}^{t-1} \alpha_i \hat{B}_i \hat{A}_i + B_t A_t, \qquad t \geq 2, \qquad (5)$$

where $W_0$ is the pretrained LLM weight; $\{\hat{B}_i\}_{i=1}^{t-1}$ and $\{\hat{A}_i\}_{i=1}^{t-1}$ are frozen adapters from previous stages; and $B_t, A_t$ are trainable at stage $t$. Following prior practice, we use normalized directions $\hat{B}_i = B_i / \|B_i\|_F$ and $\hat{A}_i = A_i / \|A_i\|_F$, which improves stability. The scalar $\alpha_i$ are fixed or learned magnitudes. This design explicitly enhances stability and expands LoRA's effective capacity, expected too work well when sequential tasks interfere minimally. However, for recommendation where user preferences evolve, this rationale weakens. To examine this, we study SumLoRA, which uses simple summation, in four variants: (i) *all*, summing all past adapters; (ii) *latest*, summing only the most recent adapter; (iii) *all+inherit*, summing all past adapters with parameter inheritance; and

Table 1: (Left) Design choices; (Right) performance gain vs. single evolving LoRA (w.r.t. NDCG@5) in different task settings on Instrument dataset.

| Method | Design choices | | | Task settings | | |
| --- | --- | --- | --- | --- | --- | --- |
| | Learnable mag. | Only latest | Param inherit | (1) User-disjoint | (2) Natural split | Diff. (1)-(2) |
| SUMLORA$_{\text{ALL}}$ | ✗ | ✗ | ✗ | $-8.13\%$ | $-26.77\%$ | $18.64\%$ |
| SUMLORA$_{\text{LATEST}}$ | ✗ | ✓ | ✗ | $-12.20\%$ | $-22.05\%$ | $9.85\%$ |
| SUMLORA$_{\text{ALL+INHERIT}}$ | ✗ | ✗ | ✓ | $-3.25\%$ | $1.57\%$ | $-4.82\%$ |
| SUMLORA$_{\text{LATEST+INHERIT}}$ | ✗ | ✓ | ✓ | $0.00\%$ | $2.36\%$ | $-2.36\%$ |
| SD-LORA$_{\text{LATEST+INHERIT}}$ | ✓ | ✓ | ✓ | $3.25\%$ | $0.79\%$ | $2.46\%$ |

(iv) *latest+inherit*, using only the latest adapter with parameter inheritance. The *all* variant corresponds to the original design of cumulative LoRA family. We also consider SD-LoRA, which extends summation with learnable magnitudes, with *all* equivalent to Wu et al. (2025). For analysis, we focus on the empirically stronger *latest+inherit*. Table 1 summarizes these design choices.

**Two settings.** We evaluate methods in the two settings derived from the same user-item interaction data of Amazon Review (Musical Instruments) dataset: **(1) Natural chronological split:** Interactions are sorted by time; a large portion (e.g., 60%) is used for pretraining (i.e., $\mathcal{D}_1$), and the remainder is divided into four equal incremental blocks, yielding $\mathcal{D}_2, \ldots, \mathcal{D}_5$. For each $\mathcal{D}_t$, we apply leave-one-out per user (second-to-last item for validation, last item for test). See Appendix C.1 for details. **(2) Pseudo user-disjoint split:** Users are randomly partitioned into disjoint sets for $\mathcal{D}_t$ ($t = 1, \ldots, 5$), with block sizes matched to the chronological split. Item order within each user's sequence is preserved. While similar users may induce some shared preferences across stages, this setting introduces relatively less cross-stage interference than the natural chronological case.

**Results.** Table 1 reports **(1)** the relative gain vs. single evolving LoRA on the user-disjoint split, **(2)** the relative gain on the chronological split, and **(3)** their difference (i.e., **(1)-(2)**). We summarize the findings: First, the **Diff.** column shows that the original cumulative design (i.e., SUMLORA$_{\text{ALL}}$) performs much worse in the natural chronological setting than in the user-disjoint setting, confirming that it is better suited for tasks with minimal interference and ill-suited for recommendation. Second, in the **Natural split**, SUMLORA$_{\text{ALL}}$ performs worst, followed by *latest*, *all+inherit*, and *latest+inherit*, suggesting that (a) aggregating all past adapters hinders adaptation, and (b) parameter inheritance is essential for gradual, proximal evolution of LoRA with respect to the previous state. Finally, SD-LORA$_{\text{LATEST+INHERIT}}$ fails to improve over fixed-magnitude SUMLORA$_{\text{LATEST+INHERIT}}$, since useful past components are entangled with stale ones, making weighting ineffective. Overall, continual recommendation requires evolving adapters with *controlled stability*, rather than rigid reuse of past ones, to capture user preference dynamics.

## 4 PROPOSED FRAMEWORK: PESO

Our design philosophy is to (1) avoid using multiple LoRA adapters, which implicitly assume task independence, and (2) preserve past knowledge in a way that supports understanding of current user behavior. Guided by this, we propose PESO (Proximally rEgularized Single evolving lOra), which maintains a single evolving LoRA adapter and regularizes each update by keeping the current adapter close to the previous one (shown in Figure 1). We begin by presenting the *quadratic proximal framework* and its theoretical implications, and then instantiate PESO with a *softmax–KL proximal* to demonstrate its practical effect.

### 4.1 SINGLE EVOLVING LORA WITH A PROXIMAL REGULARIZER

**General framework.** We maintain a single evolving LoRA and anchor each update to the previous adapter with a proximal term. Let $v_t \in \mathbb{R}^m$ denote the concatenation of all flattened LoRA $A/B$ parameters at time stage $t$. We partition coordinates into groups $g \in \{1, \ldots, G\}$ at the level of LoRA factor matrices, so that each group corresponds to the flattened parameters of either $A$ or $B$ for one LoRA-injected module (e.g., attention projections $q/k/v/o$ and MLP projection layers). We write $v^{(g)}$ for the parameters in group $g$. The overall loss function for time stage $t$ is

$$L_t = L_{\text{ce}}^{\mathcal{D}_t} + \underbrace{\frac{\lambda}{2} \sum_{g=1}^{G} \| v_t^{(g)} - v_{t-1}^{(g)} \|_{H_{t-1}^{(g)}}^2}_{\text{proximal term}}, \quad v_t \leftarrow v_{t-1} \text{ at init,} \tag{6}$$

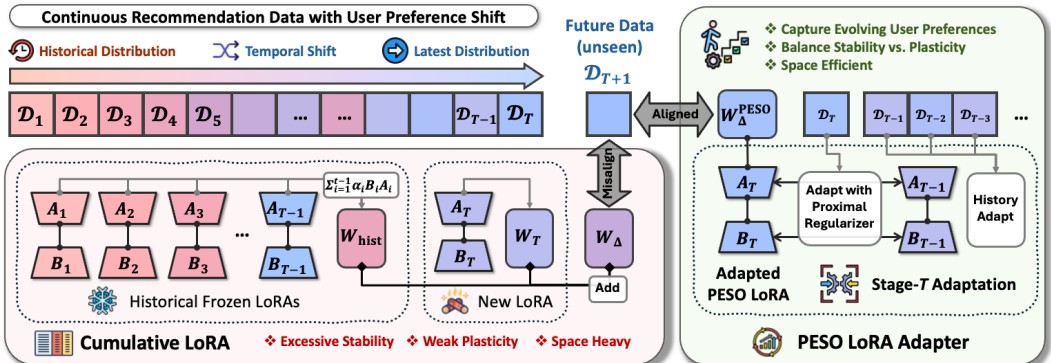

Figure 1: Overview of Cumulative LoRA and our proposed PESO with proximal regularizer.

where $L_{\text{ce}}^{\mathcal{D}_t}$ is the data-fitting term on $\mathcal{D}_t$ (i.e., cross-entropy, Eq. (2)), $\|z\|_H^2 := z^\top H z$, $\lambda > 0$ controls regularization strength, and each $H_{t-1}^{(g)} \succeq 0$ is a (symmetric) PSD metric that is fixed during stage $t$; it can be constant (e.g., $H_{t-1}^{(g)} = I$, corresponding to the L2 case) or precomputed at the previous adapter $v_{t-1}^{(g)}$. We initialize $v_t \leftarrow v_{t-1}$ so the proximal penalty starts at zero and grows only as $v_t$ departs from $v_{t-1}$. This design leverages the natural competition between the data-fitting loss (which pulls toward the optimal state for $\mathcal{D}_t$) and the proximal term (which pulls toward the previous state). Next, we theoretically show how this yields data-aware, direction-wise guidance in the LoRA subspace.

**Theoretical setup.** To analyze how the proximal term interacts with the data-fitting loss, we approximate the data-fitting term. We restrict updates to a fixed $m$-dimensional LoRA subspace. Let $\theta_0 \in \mathbb{R}^d$ be the parameter vector (base LLM and LoRA) after training on the first data block ($t=1$). From $t \geq 2$, let $\theta(v) = \theta_0 + Uv$ with $U \in \mathbb{R}^{d \times m}$ and non-LoRA coordinates frozen (i.e., assume $U = [I_m\ 0]$). For input $x = $ (prompt, item sequence) and next-item token $y$, let $s(\theta, x)$ be the scalar logit of the ground-truth token. Linearize once at $v = 0$:

$$s(\theta_0 + Uv, x) \approx s(\theta_0, x) + \Phi(x)^\top v, \qquad \Phi(x) := U^\top \nabla_\theta s(\theta_0, x) \in \mathbb{R}^m, \tag{7}$$

where $\Phi(x)$ is a tangent feature of $x$. For analysis we use a mean-squared-error surrogate for Eq. (2) and define the stage-$t$ optimum $v_t^* = \arg\min_v L^{\mathcal{D}_t}(v)$. A second-order expansion at $v_t^*$ yields quadratic loss

$$L^{\mathcal{D}_t}(v) \approx \frac{1}{2}(v - v_t^*)^\top \Sigma_t (v - v_t^*), \qquad \Sigma_t = \mathbb{E}_{x \sim \mathcal{D}_t}\big[\Phi(x)\Phi(x)^\top\big] \succeq 0, \tag{8}$$

where $\Sigma_t$ is the tangent-feature second-moment matrix for time stage $t$, capturing *how much the stage-$t$ data supports different directions in the LoRA subspace* (i.e., $u^\top \Sigma_t u = \mathbb{E}_{\mathcal{D}_t}[(\Phi(x)^\top u)^2]$ $\forall u \in \mathbb{R}^m$). See Appendix B.1 for full setup and assumptions. In what follows, we present a general proposition showing that our proximal framework yields direction-wise interpolation between the new optimum and the previous adapter, and then derive its L2 corollary to provide intuition into the stability–plasticity balance.

**Proposition 1** (Generalized–eigen interpolation with a quadratic proximal). *Let $\Sigma_t = \Sigma_t^\top \succeq 0$. Define the block-diagonal proximal metric $H_{t-1} := \text{blkdiag}\big(H_{t-1}^{(1)}, \ldots, H_{t-1}^{(G)}\big) \succeq 0$, with each $H_{t-1}^{(g)}$ symmetric PSD and independent of $v$ during stage $t$. Under the quadratic approximation in Eq. (8), our loss Eq. (6) is:*

$$L_t(v) = \frac{1}{2}(v - v_t^*)^\top \Sigma_t (v - v_t^*) + \frac{\lambda}{2}(v - v_{t-1})^\top H_{t-1}(v - v_{t-1}). \tag{9}$$

*Let $\hat{v}_t := \arg\min_v L_t(v)$. Let $\{(q_k, \rho_k)\}_{k=1}^r$ be generalized eigenpairs of $(\Sigma_t, H_{t-1})$ on $\text{range}(H_{t-1})$ (i.e., $\Sigma_t q_k = \rho_k H_{t-1} q_k$), normalized by $q_i^\top H_{t-1} q_j = \delta_{ij}$, where $r = \text{rank}(H_{t-1})$. With $\langle u, w \rangle_{H_{t-1}} := u^\top H_{t-1} w$,*

$$\langle \hat{v}_t, q_k \rangle_{H_{t-1}} = \frac{\rho_k}{\rho_k + \lambda} \langle v_t^*, q_k \rangle_{H_{t-1}} + \frac{\lambda}{\rho_k + \lambda} \langle v_{t-1}, q_k \rangle_{H_{t-1}}, \qquad k = 1, \ldots, r. \tag{10}$$

The proof of Proposition 1 is deferred to Appendix B.2. To build intuition, we specialize Proposition 1 to the *L2 case by taking* $H_{t-1} = I$. Then the generalized eigenpairs reduce to ordinary eigenpairs of $\Sigma_t$ and $\langle \cdot, \cdot \rangle_{H_{t-1}}$ becomes the standard inner product, yielding the following corollary.

**Corollary 2** (L2 special case of Proposition 1). *Take* $H_{t-1} = I$. *If* $\Sigma_t q_k = \sigma_k^2 q_k$ *with* $\{q_k\}$ *orthonormal,*

$$\langle \hat{v}_t, q_k \rangle = \frac{\sigma_k^2}{\sigma_k^2 + \lambda} \langle v_t^*, q_k \rangle + \frac{\lambda}{\sigma_k^2 + \lambda} \langle v_{t-1}, q_k \rangle, \qquad k = 1, \cdots, m. \tag{11}$$

In a nutshell, Corollary 2 shows a **data-aware balance between stability and plasticity** in our framework. Recall that $\Sigma_t = \mathbb{E}_{\mathcal{D}_t}[\Phi(x)\Phi(x)^\top]$ summarizes how much the stage-$t$ data *supports* different directions in the LoRA subspace. Its eigenvectors $q_k$ are principal directions, with eigenvalues $\sigma_k^2$ measuring the strength of support along each direction under $\mathcal{D}_t$. By Corollary 2, along any $q_k$ the update is a weighted average of $v_t^*$ and $v_{t-1}$, with weight toward $v_t^*$ equal to $\sigma_k^2/(\sigma_k^2 + \lambda)$. Thus, when $\sigma_k^2$ is large (strong support in $\mathcal{D}_t$), $\hat{v}_t$ moves toward $v_t^*$ along $q_k$ (e.g., the user starts engaging more with mystery than sci-fi); when $\sigma_k^2$ is small (weak support), $\hat{v}_t$ stays close to $v_{t-1}$ (e.g., a stable brand affinity not observed this week). If $\sigma_k^2 = 0$, the component along $q_k$ is kept exactly from the previous stage. See Appendix B.3 for a more detailed intuitive explanation.

## 4.2 SOFTMAX–KL AS A PROXIMAL REGULARIZER

As shown earlier, the L2 proximal (i.e., $H_{t-1}^{(g)} = I$) is a special case of our general proximal form with $H_{t-1}$. However, it penalizes all coordinate changes equally, treating modules uniformly, ignoring internal structure, and not adapting to the previous state $v_{t-1}$. To address this, we instantiate the proximal term with a *softmax–KL proximal* that preserves per-module structure and leverages the previous state. Formally, the stage-$t$ objective of PESO is:

$$L_t = L_{\text{ce}}^{\mathcal{D}_t} + \lambda \underbrace{\sum_{g=1}^{G} D_{\text{KL}}\big(\text{softmax}(v_t^{(g)}) \,\|\, \text{softmax}(v_{t-1}^{(g)})\big)}_{\mathcal{K}_{\text{blk}}(v_t, v_{t-1})}, \quad v_t \leftarrow v_{t-1} \text{ at init.} \tag{12}$$

We first show that the softmax–KL proximal locally reduces to a quadratic form, and then give a corollary that interprets it as a $p$-weighted variance, providing an intuitive view of its module-wise stability.

**Proposition 3** (Per-module softmax–KL is locally quadratic). *Let* $v_t^{(g)}$ *be the subvector for group* $g \in \{1, \ldots, G\}$ *(e.g., LoRA A/B of a module),* $p^{(g)} = \text{softmax}(v_{t-1}^{(g)})$, *and* $\Delta^{(g)} = v_t^{(g)} - v_{t-1}^{(g)}$. *For small* $\Delta^{(g)}$,

$$\mathcal{K}_{\text{blk}}(v_t, v_{t-1}) = \frac{\lambda}{2} \sum_{g=1}^{G} \big(\Delta^{(g)}\big)^\top \Big(\text{diag}\big(p^{(g)}\big) - p^{(g)} p^{(g)^\top}\Big) \Delta^{(g)} + o(\|\Delta\|^2) \tag{13}$$

$$= \frac{\lambda}{2} \Delta^\top \underbrace{\text{blkdiag}\big(H_{t-1}^{(1)}, \ldots, H_{t-1}^{(G)}\big)}_{=:H_{t-1}} \Delta + o(\|\Delta\|^2), \text{ with } H_{t-1}^{(g)} = \text{diag}(p^{(g)}) - p^{(g)}(p^{(g)})^\top \succeq 0.$$

The proof of Proposition 3 is deferred to Appendix B.4. Proposition 3 shows the **softmax–KL proximal is locally equivalent to the quadratic form** $\frac{\lambda}{2}\|v_t - v_{t-1}\|_{H_{t-1}}^2$ with $H_{t-1} = \text{blkdiag}(H_{t-1}^{(1)}, \ldots, H_{t-1}^{(G)})$. Hence, Proposition 1 applies directly, suggesting a data-aware balance of stability and plasticity.

**Corollary 4** (Softmax–KL equals $p$-weighted variance). *With notation as above, up to an additive constant,*

$$\mathcal{K}_{\text{blk}}(v_t, v_{t-1}) = \frac{\lambda}{2} \sum_{g=1}^{G} \text{Var}_{p^{(g)}}\big(\Delta^{(g)}\big), \tag{14}$$

*where* $\text{Var}_{p^{(g)}}(\Delta^{(g)}) = \sum_{i \in g} p_i^{(g)} (\Delta_i^{(g)} - \mu^{(g)})^2$ *and* $\mu^{(g)} = \sum_{i \in g} p_i^{(g)} \Delta_i^{(g)}$.

Corollary 4 shows that the softmax–KL proximal can be interpreted as a *p-weighted variance* of parameter changes. Consequently, it penalizes (1) *reshuffling* (i.e., relative (centered) changes) within each module's LoRA factor, and (2) deviations more strongly for coordinates with higher prior mass. **This yields module-wise, previous-state–aware stability** without killing plasticity: updates still move toward new optima where data provides strong support (as in Proposition 1), while staying close to the previous state otherwise.

## 5 EXPERIMENTS

We design experiments to answer: **RQ1:** To what extent does PESO outperform competitors? **RQ2:** Which proximal regularizer works best in PESO? **RQ3**: How effectively does PESO balance stability and plasticity under different user drift patterns? **RQ4:** How do hyperparameters affect performance of PESO? **RQ5**: How does PESO compare to traditional continual recommenders?

*Please refer to Appendix C for additional experimental results and analyses, including distribution-shift quantification, comparisons with additional baselines (full-parameter fine-tuning, O-LoRA, AM-LoRA, LSAT etc.), experiments with other backbones and datasets, and an efficiency analysis.*

### 5.1 EXPERIMENTAL SETTINGS

**Datasets.** We use the real-world Amazon Review dataset, which contains user reviews (treated as implicit interactions) on products over time. We focus on three categories: Musical Instruments, Movies & TV, and Books. Detailed preprocessing steps and dataset statistics are provided in Appendix C.1. The processed data yield $\{\mathcal{D}_1, \ldots, \mathcal{D}_5\}$, where $\mathcal{D}_1$ is a large pretraining set and $\mathcal{D}_2, \ldots, \mathcal{D}_4$ are smaller incremental sets.

**Evaluation.** For each $\mathcal{D}_t$, we apply leave-one-out evaluation per user, reserving the last item for testing. Following (Wang et al., 2024; Bao et al., 2025a), we construct multiple training pairs $(x_u, y_u)$ per user using a sliding window of size 20. Starting from the LLM pretrained on $\mathcal{D}_1$, at each stage $t = 2, \ldots, 5$ the model is fine-tuned and then generates 10 items via constrained beam search restricted to valid item tokens. We report Hit@5/10 and NDCG@5/10, averaged over $\mathcal{D}_2, \ldots, \mathcal{D}_4$. Full evaluation details are in Appendix C.1.

**Compared methods and implementation details.** We compare PESO with several LoRA-based baselines for continual learning, all using the same cross-entropy loss and Llama-3.2 1B (Grattafiori et al., 2024) as backbone. The bottom baseline is PRETRAIN, trained on $\mathcal{D}_1$ and directly evaluated at $t = 2, \ldots, 4$. Among continual methods, we consider: (1) *single evolving LoRA*; and (2) the *cumulative family*, which combines past and current adapters: *SumLoRA*, *SD-LoRA* (Wu et al., 2025), and *InfLoRA* (Liang & Li, 2024). SD-LoRA learns magnitudes for normalized past adapters, while InfLoRA precomputes LoRA-$A$ via SVD of the input covariance and trains only $B$, to better align with current data and reduce task-interference. As discussed in Section 3, original cumulative designs use *all past adapters without inheritance* (*all*). For recommendation, we further test three variants: *latest* (most recent only), *all+inherit* (all with inheritance), and *latest+inherit* (latest with inheritance). For hyperparameters, $\lambda$ is searched over $[0.5, 1.0, 2.0, 5.0, 8.0]$ (set to 2.0 for Instruments, 5.0 for Movies&TV and Books). SD-LoRA magnitudes start at 1.0.

### 5.2 EXPERIMENTAL RESULTS AND DISCUSSION

**Main Results (RQ1).** Table 2 reports results across four metrics and three datasets in continual settings. First, all continual learning methods consistently outperform PRETRAIN, highlighting the importance of adapting to new data to capture evolving user preferences, even when incremental data is much smaller (e.g., 10%) than the pretraining data. Second, neither single evolving LoRA nor the cumulative family dominates, while PESO consistently achieves the best results, with average gains of 3.71%, 4.62%, and 6.26% over the best competitors (SINGLE EVOLVING LoRA, SUMLORA_LATEST+INHERIT, and SD-LORA_LATEST+INHERIT). Cumulative LoRA, though more complex and storage-heavy, often underperforms or only matches single evolving LoRA, as rigidly reusing frozen adapters overly constrains adaptation to evolving user preferences. By contrast, PESO uses flexible proximal regularization toward the latest state, allowing the data-fitting loss and proximal term to jointly decide what to preserve or update. Third, as discussed in detail in Section 3, regarding SumLoRA and SD-LoRA, original cumulative designs (using all past adapters without inheritance) perform worst, while variants with inheritance or only the latest adapter do better. Notably,

Table 2: Recommendation performance averaged across time stages for PESO and continual competitors. The best and second-best results are marked in **bold** and underline, respectively.

| Methods | Instruments | | | | Movies & TVs | | | | Books | | | |
|---|---|---|---|---|---|---|---|---|---|---|---|---|
| | H@5 | H@10 | N@5 | N@10 | H@5 | H@10 | N@5 | N@10 | H@5 | H@10 | N@5 | N@10 |
| PRETRAIN | 0.0166 | 0.0216 | 0.0115 | 0.0131 | 0.0166 | 0.0231 | 0.0111 | 0.0132 | 0.0258 | 0.0283 | 0.0196 | 0.0204 |
| SINGLE EVOLVING LORA | 0.0181 | 0.0253 | 0.0127 | 0.0150 | 0.0175 | 0.0247 | 0.0116 | 0.0138 | **0.0448** | 0.0557 | 0.0308 | 0.0344 |
| Cumulative LoRA Family | | | | | | | | | | | | |
| INFLORA$_{ALL}$ | 0.0156 | 0.0214 | 0.0105 | 0.0124 | 0.0103 | 0.0139 | 0.0067 | 0.0079 | 0.0236 | 0.0332 | 0.0161 | 0.0193 |
| INFLORA$_{LATEST}$ | 0.0131 | 0.0167 | 0.0090 | 0.0102 | 0.0073 | 0.0092 | 0.0047 | 0.0054 | 0.0152 | 0.0197 | 0.0108 | 0.0123 |
| INFLORA$_{ALL+INHERIT}$ | 0.0149 | 0.0219 | 0.0104 | 0.0126 | 0.0109 | 0.0147 | 0.0072 | 0.0085 | 0.0249 | 0.0324 | 0.0171 | 0.0195 |
| INFLORA$_{LATEST+INHERIT}$ | 0.0137 | 0.0202 | 0.0095 | 0.0116 | 0.0094 | 0.0132 | 0.0060 | 0.0072 | 0.0225 | 0.0288 | 0.0153 | 0.0174 |
| SUMLORA$_{ALL}$ | 0.0134 | 0.0215 | 0.0093 | 0.0119 | 0.0102 | 0.0130 | 0.0067 | 0.0076 | 0.0264 | 0.0402 | 0.0176 | 0.0221 |
| SUMLORA$_{LATEST}$ | 0.0143 | 0.0221 | 0.0099 | 0.0124 | 0.0102 | 0.0130 | 0.0067 | 0.0076 | 0.0246 | 0.0354 | 0.0161 | 0.0196 |
| SUMLORA$_{ALL+INHERIT}$ | 0.0182 | 0.0260 | 0.0129 | 0.0154 | 0.0160 | 0.0234 | 0.0107 | 0.0131 | 0.0409 | 0.0514 | 0.0287 | 0.0321 |
| SUMLORA$_{LATEST+INHERIT}$ | 0.0185 | 0.0255 | 0.0130 | 0.0152 | 0.0172 | 0.0237 | 0.0114 | 0.0135 | 0.0433 | 0.0542 | 0.0306 | 0.0341 |
| SD-LORA$_{ALL}$ | 0.0156 | 0.0226 | 0.0107 | 0.0129 | 0.0094 | 0.0133 | 0.0061 | 0.0074 | 0.0238 | 0.0351 | 0.0162 | 0.0198 |
| SD-LORA$_{LATEST}$ | 0.0156 | 0.0218 | 0.0102 | 0.0123 | 0.0101 | 0.0142 | 0.0069 | 0.0082 | 0.0241 | 0.0327 | 0.0159 | 0.0186 |
| SD-LORA$_{ALL+INHERIT}$ | 0.0176 | 0.0238 | 0.0124 | 0.0144 | 0.0118 | 0.0171 | 0.0077 | 0.0094 | 0.0332 | 0.0412 | 0.0234 | 0.0260 |
| SD-LORA$_{LATEST+INHERIT}$ | 0.0184 | 0.0254 | 0.0128 | 0.0150 | 0.0165 | 0.0235 | 0.0109 | 0.0131 | 0.0432 | 0.0530 | 0.0308 | 0.0340 |
| **PESO** | **0.0193** | **0.0268** | **0.0138** | **0.0162** | **0.0180** | **0.0251** | **0.0118** | **0.0141** | **0.0448** | **0.0569** | **0.0311** | **0.0351** |
| Performance Gain (%) | | | | | | | | | | | | |
| VS. SINGLE EVOLVING LORA | 6.63% | 5.93% | 8.66% | 8.00% | 2.86% | 1.62% | 1.72% | 2.17% | 0.00% | 2.15% | 0.97% | 2.03% |
| VS. SUMLORA$_{LATEST+INHERIT}$ | 4.32% | 5.10% | 6.15% | 6.58% | 4.65% | 5.91% | 3.51% | 4.44% | 3.46% | 4.98% | 1.63% | 2.93% |
| VS. SD-LORA$_{LATEST+INHERIT}$ | 4.89% | 5.51% | 7.81% | 8.00% | 9.09% | 6.81% | 8.26% | 7.63% | 3.70% | 7.36% | 0.97% | 3.24% |

Figure 2: Comparison of different regularization methods against the previous LoRA.

Figure 3: Impact of the scaling weight $\lambda$ for the proximal term on PESO performance.

some non-inheritance variants even fall below PRETRAIN, showing that without gradual evolution, continual learning can harm more than help. InfLoRA yields the weakest results overall, likely because, although it incorporates input data covariance information, freezing $A$ prevents inheritance and gradual adaptation across time, both of which are crucial in continual recommendation.

**Analysis on Proximal Regularizer (RQ2).** Unless otherwise noted, all subsequent subsections report average performance across four metrics (Hit@5, Hit@10, NDCG@5, NDCG@10). We compare PESO with four alternative regularizers on the previous adapter: orthogonality, L2 proximal, LoRA-Output KL, and Per-Rank KL (Figure 2). Orthogonality, an interference-minimization strategy common in vision, performs far worse than all methods, showing that minimizing interference across stages is harmful in continual recommendation. L2 proximal, which penalizes the L2 distance between current and previous parameters, is often comparable to single evolving LoRA but worse than PESO, suggesting that uniform constraints are insufficient. LoRA-Output KL (softmax-KL applied in LoRA output, i.e., function space) and Per-Rank KL (softmax-KL applied on each rank of LoRA matrcies, i.e., finer parameter granularity) are slightly worse or comparable to PESO, suggesting that regularization directly in the parameter space with module-aware structure is more effective, or at least sufficient, compared to output-level or overly fine-grained constraints.

**Stability–Plasticity Analysis via User Groups (RQ3).** To examine how PESO balances long-term interests with newly evolved preferences, we evaluate the final model on two user groups in the Instruments dataset, which serve as proxies for different drift patterns: (1) **Dormant Users:** Users who were active

Table 3: Performance (NDCG@5) across user groups representing stability and plasticity tests.

| Method | Dormant Users | New Users |
|---|---|---|
| Single Evolving LoRA (Plasticity) | 0.0154 | 0.0116 |
| Cumulative LoRA (Stability) | 0.0164 | 0.0101 |
| **PESO (Balanced)** | **0.0170** | **0.0122** |

in earlier blocks, absent in intermediate blocks, and return in $D_4$. This tests **stability** (retention of long-term preferences). (2) **New Users:** Users who appear only in $D_4$. This tests **plasticity** (adaptation to new signals). Table 3 illustrates the trade-off: Single Evolving LoRA excels on New Users but performs poorly on Dormant Users due to forgetting, whereas Cumulative LoRA (i.e., SUMLORA$_{LATEST+INHERIT}$) preserves stability but adapts less effectively to New Users. PESO achieves the best performance on both groups, indicating a strong stability–plasticity balance.

**Hyperparameter Analysis (RQ4). (a) Scaling parameter $\lambda$ for proximal term in PESO.** Figure 3 shows performance as $\lambda$ varies. Starting from $\lambda = 0$ (i.e., single evolving LoRA), performance

improves as $\lambda$ increases, then either decreases or plateaus, confirming that $\lambda$ serves as a tunable trade-off between stability and plasticity: too small harms stability; too large harms plasticity. In addition, performance is not highly sensitive to $\lambda$, as results remain stable across a broad range of values. **(b) Learning rate for continual stages.** See Appendix C.2 for full results and discussion. Since incremental datasets are much smaller than the pretraining set, performance is highly sensitive to learning rate. Our results show that using the pretraining rate leads to overfitting, while scaling the rate down ($\approx 0.05$–$0.1\times$) yields the best performance.

**Comparison with Traditional Continual Recommenders (RQ5).** Details are in Appendix C.3; Table 4 shows a subset (top: traditional, bottom: LLM-based). LLM-based methods generally outperform traditional two-tower models, except on Instruments, where explicit dual modeling of users and items helps. While PESO achieves higher absolute performance, continual methods like PISA (Yoo et al., 2025a) yield larger relative gains in two-tower models, reflecting the advantage of explicit user embeddings in capturing preference drift and the challenge of doing so with LLMs.

Table 4: Comparison of traditional and LLM-based methods.

| Method | Instruments | Movies & TVs | Books |
|---|---|---|---|
| Pretrain | 0.0153 | 0.0028 | 0.0041 |
| Fine-tuning | 0.0180 | 0.0114 | 0.0218 |
| PISA | 0.0194 | 0.0106 | 0.0301 |
| Pretrain | 0.0157 | 0.0160 | 0.0235 |
| Fine-tuning | 0.0178 | 0.0169 | 0.0414 |
| PESO | 0.0190 | 0.0173 | 0.0422 |

## 6 RELATED WORKS

**LLM-based Generative Recommender Systems.** Recent advances in large language models (LLMs) have inspired generative approaches to recommendation, where the task is framed as sequence generation. Instead of ranking items from a candidate set, the model autoregressively generates the next item token given a user's interaction history. Variants of this paradigm includes zero-shot prompting (Lyu et al., 2023), ID-token generation (Tan et al., 2024; Wang et al., 2024), data-efficient fine-tuning (Lin et al., 2024b), uncertainty-aware decoding (Kweon et al., 2025), and alignment techniques for recommendation objectives (Cao et al., 2024; Bao et al., 2025a; Chen et al., 2024). These works demonstrate that LLMs can flexibly leverage textual and structural signals for recommendation, but they typically assume static data. In contrast, real-world interactions arrive continuously, requiring models that can adapt to evolving user preferences without costly retraining. Our work addresses this gap by studying continual adaptation of generative LLM recommenders.

**Continual Learning for Foundational Models and LoRA.** Classical continual recommenders use parameter regularization (Wang et al., 2021; 2023b; Yoo et al., 2024), replay buffers (Ahrabian et al., 2021; Zhang et al., 2024b; Zhu et al., 2023; Zou et al., 2025), or dynamic architectures (He et al., 2023; Zhang et al., 2023; Lee et al., 2025b). With large foundational models, parameter-efficient fine-tuning (PEFT) has become central, with LoRA (Hu et al., 2022) as a standard choice. In vision, several continual extensions have been proposed, such as cumulative aggregation of frozen adapters (Liang & Li, 2024; Lu et al., 2024) and learnable magnitude scaling (SD-LoRA) (Wu et al., 2025), which are effective when tasks interference is minimal. However, these methods are less suitable for recommendation, where user preferences evolve over time. Our work differs by proposing a proximal single evolving LoRA that avoids the forgetting of single evolving LoRA and the rigidity of cumulative LoRA, better suiting the continual recommendation setting.

## 7 CONCLUSION

We have studied the problem of continual adaptation for LLM-based generative recommender systems, where user interactions arrive over time and preferences evolve. Single evolving LoRA offers strong plasticity but suffers from forgetting, while cumulative LoRA improves stability but entangles outdated signals. Our proposed PESO strikes a better balance by maintaining a single adapter and regularizing it toward its prior state, allowing the model to decide what to adapt and what to preserve. Our theoretical analysis has shown that the proximal design provides data-aware, direction-wise guidance in the LoRA subspace, and our instantiation with per-module softmax–KL further preserves internal parameter structure. Empirical results across multiple real-world datasets confirm that PESO consistently outperforms existing baselines, achieving a superior stability–plasticity balance. Future directions include drift-aware and more personalized regularization, as well as integrating our approach with other parameter-efficient tuning methods to further improve continual adaptation in large foundation models.

ACKNOWLEDGEMENTS

This work is supported by NSF (2324770) and AFOSR (FA9550-24-1-0002). The content of the information in this document does not necessarily reflect the position or the policy of the Government, and no official endorsement should be inferred. The U.S. Government is authorized to reproduce and distribute reprints for Government purposes notwithstanding any copyright notation here on.

ETHICS STATEMENT.

This work focuses on continual learning methods for large language model (LLM)-based recommender systems. It does not involve human subjects, sensitive personal data, or private user information. All experiments are conducted on publicly available benchmark datasets (Amazon Reviews). We followed standard preprocessing protocols, and no personally identifiable information was used or released. While recommender systems can influence user exposure to content, this study is purely methodological and does not deploy or interact with real users. We acknowledge the potential societal risks of recommendation technologies, such as reinforcing biases or filter bubbles, and we emphasize that our method (PESO) is designed as a modular continual learning technique, independent of any particular application domain or societal factors.

REPRODUCIBILITY STATEMENT.

The paper provides: (1) detailed descriptions of datasets, preprocessing steps, and evaluation protocols (Section 5.1, Appendix C.1); (2) clear definitions of baselines, the proposed method (PESO), and its theoretical analyses (Sections 3, 4, Appendix B); and (3) hyperparameter settings, search ranges, and sensitivity analyses (Section 5). Results are reported across multiple datasets and metrics for robustness. Full proofs are included in Appendix B. We will release our implementation and data-processing scripts upon publication to ensure reproducibility.

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

## A    CONCEPTUAL MODELING OF EVOLVING USER PREFERENCES

We assume an initial model is pretrained offline on base data $\mathcal{D}_1$, and then fine-tuned sequentially on chronologically arriving blocks $\mathcal{D}_2, \ldots, \mathcal{D}_T$. Let $x_u^{t-1}$ denote $u$'s interaction history available before stage $t$, and let $P_t(y \mid x_u^{t-1})$ be the conditional distribution of the next item $y$ during stage $t$, representing user preferences. In continual recommendation, these distributions evolve over time, which can be conceptually modeled as

$$P_t(y \mid x_u^{t-1}) \approx \alpha_t \, P_{t-1}(y \mid x_u^{t-1}) \, + \, (1 - \alpha_t) \, Q_t(y \mid x_u^{t-1}), \tag{15}$$

where $P_{t-1}$ captures stability (persistent long-term preferences), $Q_t$ captures plasticity (new or shifting preferences estimated from new data), and $\alpha_t \in [0, 1]$ controls the balance. The goal is to minimize expected risk on upcoming interactions by balancing stability and plasticity.

## B    DETAILED THEORETICAL ANALYSIS

### B.1    SETUP AND ASSUMPTIONS

**Assumption 1** (Parameters and LoRA subspace). *Let $\theta \in \mathbb{R}^d$ denote the vectorized concatenation of all model parameters (base LLM and LoRA). Let $\theta_0$ be the parameter vector after training on the first data block (t=1). From $t \geq 2$, restrict updates to a fixed $m$-dimensional LoRA subspace spanned by columns of $U \in \mathbb{R}^{d \times m}$ and write*

$$\theta = \theta_0 + Uv, \qquad v \in \mathbb{R}^m, \tag{16}$$

*with all non-LoRA coordinates frozen. Without loss of generality, assume $U = [I_m \ 0]$, i.e., the LoRA subspace is the first $m$ coordinates.*

**Assumption 2** (Linearization and tangent features.). *Let $s(\theta, x) \in \mathbb{R}$ be the scalar logit of the ground-truth next item. We linearize $s$ at $v = 0$ (i.e., at $\theta = \theta_0$):*

$$s(\theta_0 + Uv, x) \approx s(\theta_0, x) + v^\top U^\top \nabla_\theta s(\theta_0, x) = s_0(x) + v^\top \Phi(x), \tag{17}$$

*with tangent features of $x$*

$$\Phi(x) := U^\top \nabla_\theta s(\theta_0, x) \in \mathbb{R}^m. \tag{18}$$

**Assumption 3** (Data and loss.). *Let $(x, y) \sim \mathcal{D}_t$ be examples in block $t$. In recommendation, $x = (\text{prompt, item sequence})$ and $y \in \mathcal{V}$ is the next-item token. Training typically uses cross-entropy on logits; for analysis, we use a mean-squared-error (MSE) surrogate. Define the block-$t$ risk*

$$L^{\mathcal{D}_t}(v) = \mathbb{E}_{(x,y) \sim \mathcal{D}_t} \left[ \tfrac{1}{2} \big(s(\theta_0 + Uv, x) - r_t(x, y)\big)^2 \right]. \tag{19}$$

*where $r_t(x, y) \in \mathbb{R}$ is a calibrated target score for the ground-truth next item.*

Note that under the linearization, this yields a quadratic risk with positive-semidefinite curvature. All later proofs use only this PSD curvature, not the exact form of $r_t$.

**Assumption 4** (Quadratic form under the linearization.). *Substituting $s(\theta_0 + v, x) \approx s_0(x) + \Phi(x)^\top v$ gives, up to an additive constant,*

$$L^{\mathcal{D}_t}(v) = b_t^\top v + \tfrac{1}{2} v^\top \Sigma_t v, \qquad b_t := \mathbb{E}_{\mathcal{D}_t}\big[(s_0(x) - r_t(x, y)) \, \Phi(x)\big], \quad \Sigma_t := \mathbb{E}_{\mathcal{D}_t}\big[\Phi(x) \Phi(x)^\top\big] \succeq 0. \tag{20}$$

*Define stage-$t$ optimum $v_t^*$ as any minimizer of $L^{\mathcal{D}_t}(v)$*

$$v_t^* = \arg \min_v L^{\mathcal{D}_t}(v). \tag{21}$$

*A second-order Taylor expansion of $L_t^{\mathcal{D}_t}$ at $v_t^*$ gives*

$$L^{\mathcal{D}_t}(v) = L^{\mathcal{D}_t}(v_t^*) + \underbrace{(\nabla L^{\mathcal{D}_t})^\top(v_t^*)}_{=0}(v - v_t^*) + \tfrac{1}{2}(v - v_t^*)^\top \underbrace{\nabla^2 L^{\mathcal{D}_t}(v_t^*)}_{=\Sigma_t}(v - v_t^*). \tag{22}$$

*Dropping the constant term, the centered quadratic risk used throughout is*

$$L^{\mathcal{D}_t}(v) = \tfrac{1}{2}(v - v_t^*)^\top \Sigma_t (v - v_t^*), \tag{23}$$

where $\Sigma_t$ is the tangent-feature second-moment matrix for time stage $t$, capturing *how much the stage-t data supports different directions in the LoRA subspace* (i.e., $u^\top \Sigma_t u = \mathbb{E}_{\mathcal{D}_t}[(\Phi(x)^\top u)^2]$ $\forall u \in \mathbb{R}^m$). Also note that we fix the linearization at $\theta_0$: $s(\theta_0 + Uv, x) \approx s_0(x) + \Phi(x)^\top v$ with $\Phi(x) = U^\top \nabla_\theta s(\theta_0, x)$. Although $\Phi(x)$ is fixed across $t$, the $\Sigma_t = \mathbb{E}_{\mathcal{D}_t}[\Phi(x)\Phi(x)^\top]$ varies with the data block distribution, so drift is captured via $\Sigma_t$ and the shifting optimum $v_t^*$.

**Remark: relinearization per block.** If desired, one may instead relinearize at $\theta_0 + Uv_{t-1}$, replacing $\Phi(x)$ by $\Phi_{t-1}(x) = U^\top \nabla_\theta s(\theta_0 + Uv_{t-1}, x)$ and $\Sigma_t$ by $\mathbb{E}[\Phi_{t-1}\Phi_{t-1}^\top]$. All propositions and closed forms carry over with these substitutions; the only change is that the curvature reflects the anchor $v_{t-1}$ of the current block. We found fixed linearization sufficient and notationally lighter.

## B.2 PROOF OF PROPOSITION 1.

**Assumption 5** (Complementarity (no doubly–flat directions)). *On the LoRA subspace $\mathbb{R}^m$, let $\Sigma_t \succeq 0$ and $H_{t-1} \succeq 0$ be symmetric and fixed with respect to $v$. Assume*

$$\ker(\Sigma_t) \cap \ker(H_{t-1}) = \{0\}. \tag{24}$$

*Equivalently, for any $x \in \mathbb{R}^m \setminus \{0\}$, $x^\top \Sigma_t x > 0$ or $x^\top H_{t-1} x > 0$.*

*Proof.* Recall that $\hat{v}_t$ denotes the minimizer of $L_t(v)$ in Eq. (9).

**(i) Closed form of the minimizer.** Differentiating yields

$$\nabla_v L_t(v) = \Sigma_t(v - v_t^*) + \lambda H_{t-1}(v - v_{t-1}). \tag{25}$$

Setting $\nabla_v L_t(\hat{v}_t) = 0$ gives the normal equation

$$(\Sigma_t + \lambda H_{t-1})\,\hat{v}_t = \Sigma_t v_t^* + \lambda H_{t-1}v_{t-1}. \tag{26}$$

Under Assumption 5, for any $x \neq 0$,

$$x^\top (\Sigma_t + \lambda H_{t-1})x = x^\top \Sigma_t x + \lambda x^\top H_{t-1}x > 0, \tag{27}$$

so $\Sigma_t + \lambda H_{t-1} \succ 0$ and Eq. (26) has the unique solution

$$\hat{v}_t = (\Sigma_t + \lambda H_{t-1})^{-1}\big(\Sigma_t v_t^* + \lambda H_{t-1}v_{t-1}\big). \tag{28}$$

**(ii) Direction-wise interpolation in a generalized eigenbasis.** Let $\{(q_k, \rho_k)\}_{k=1}^r$ be generalized eigenpairs of $(\Sigma_t, H_{t-1})$ on $\mathrm{range}(H_{t-1})$, i.e., $\Sigma_t q_k = \rho_k H_{t-1} q_k$, normalized by $q_i^\top H_{t-1} q_j = \delta_{ij}$. Left-multiply Eq. (26) by $q_k^\top$ to obtain

$$q_k^\top \Sigma_t \hat{v}_t + \lambda q_k^\top H_{t-1} \hat{v}_t = q_k^\top \Sigma_t v_t^* + \lambda q_k^\top H_{t-1} v_{t-1}. \tag{29}$$

Using the symmetry of $\Sigma_t$ and $H_{t-1}$, together with $\Sigma_t q_k = \rho_k H_{t-1} q_k$ and $\langle u, w \rangle_{H_{t-1}} := u^\top H_{t-1} w$, we obtain

$$(\rho_k + \lambda)\langle \hat{v}_t, q_k \rangle_{H_{t-1}} = \rho_k \langle v_t^*, q_k \rangle_{H_{t-1}} + \lambda \langle v_{t-1}, q_k \rangle_{H_{t-1}}. \tag{30}$$

Dividing by $\rho_k + \lambda$ yields the claimed interpolation for $k = 1, \ldots, r$.

**(iii) Note on directions in** $\ker(H_{t-1})$**.** If $H_{t-1} \succ 0$, then $r = m$ and the above holds for all directions. If $H_{t-1}$ is singular, the statement is posed on $\mathrm{range}(H_{t-1})$; uniqueness of $\hat{v}_t$ is still guaranteed by Assumption 5, which rules out doubly-flat directions in $\ker(\Sigma_t) \cap \ker(H_{t-1})$. $\square$

## B.3 INTUITIVE EXPLANATION OF COROLLARY 2

Here, we provide an intuitive explanation of how PESO offers data-aware, direction-wise guidance.

- **Semantics of directions ($q_k$): decoupled preference axes.** The eigenvectors $q_k$ of the tangent-feature second-moment matrix $\Sigma_t$ represent principal directions in the LoRA subspace. Intuitively, these directions can be viewed as approximately independent latent axes of user behavior (e.g., one axis may correspond to "sci-fi affinity" while another reflects "price sensitivity"). Since the $q_k$'s form an orthogonal basis (in the L2 case), PESO can adjust the model along one axis with minimal interference to others, helping preserve long-term knowledge while adapting to new signals.

- **Mechanism ($\sigma_k^2$): signal strength as a gate.** The eigenvalue $\sigma_k^2$ measures how strongly the current block $\mathcal{D}_t$ supports updates along direction $q_k$. Corollary 2 shows that PESO interpolates between the new-data optimum $v_t^*$ and the previous adapter $v_{t-1}$ with weight $\sigma_k^2/(\sigma_k^2 + \lambda)$:
    - **Large $\sigma_k^2$ (strong support $\Rightarrow$ plasticity).** When $\mathcal{D}_t$ provides strong evidence along a direction (e.g., the user recently engages heavily with mystery content), the model is encouraged to move toward $v_t^*$ along $q_k$, enabling rapid adaptation to evolving interests.
    - **Small $\sigma_k^2$ (weak support $\Rightarrow$ stability).** When $\mathcal{D}_t$ provides little evidence along a direction (e.g., a stable preference that does not appear in recent interactions), updates along $q_k$ are down-weighted and the solution stays close to $v_{t-1}$, preventing long-term interests from being overwritten by weak or noisy signals.

### B.4 PROOF OF PROPOSITION 3

To prove Proposition 3, we first establish the following proposition for arbitrary $v_t$ and $v_{t-1}$, and then extend it to the blockwise case.

**Proposition 5** (Local quadratic form of softmax-KL proximal). *Let $p := \mathrm{softmax}(v_{t-1}) \in \mathbb{R}^n$ and $\Delta := v_t - v_{t-1}$. Define*

$$\mathcal{K}(\Delta) := D_{\mathrm{KL}}\big(\mathrm{softmax}(v_{t-1} + \Delta) \,\|\, \mathrm{softmax}(v_{t-1})\big). \tag{31}$$

*Then $\mathcal{K}(0) = 0$, $\nabla\mathcal{K}(0) = 0$, and the second-order Taylor expansion at $\Delta = 0$ is*

$$\mathcal{K}(\Delta) = \tfrac{1}{2}\,\Delta^\top\big(\mathrm{diag}(p) - pp^\top\big)\,\Delta + o(\|\Delta\|^2). \tag{32}$$

*Equivalently,*

$$\mathcal{K}(\Delta) = \tfrac{1}{2}\,\underbrace{\Big(\sum_{i=1}^n p_i\,(\Delta_i - \mu)^2\Big)}_{\mathrm{Var}_p(\Delta)} + o(\|\Delta\|^2), \qquad \mu := \sum_{i=1}^n p_i\,\Delta_i. \tag{33}$$

*Proof.* Write $r(\Delta) := \mathrm{softmax}(v_{t-1} + \Delta) \in \mathbb{R}^n$ and $p := r(0) = \mathrm{softmax}(v_{t-1})$. By definition,

$$\mathcal{K}(\Delta) = \sum_{i=1}^n r_i(\Delta) \log\frac{r_i(\Delta)}{p_i}. \tag{34}$$

(i) At $\Delta = 0$ we have $r(0) = p$, so

$$\mathcal{K}(0) = \sum_i p_i \log(p_i/p_i) = 0. \tag{35}$$

For the gradient, differentiate using the scalar identity $\frac{d}{dx}[x\log(x/c)] = \log(x/c) + 1$:

$$\frac{\partial\mathcal{K}}{\partial\Delta_a} = \sum_{i=1}^n \frac{\partial r_i}{\partial\Delta_a}\left(\log\frac{r_i}{p_i} + 1\right). \tag{36}$$

Evaluating at $\Delta = 0$ gives $\log(r_i/p_i) = 0$ and hence

$$\Big[\nabla\mathcal{K}(0)\Big]_a = \sum_{i=1}^n \Big[\frac{\partial r_i}{\partial\Delta_a}\Big]_{\Delta=0} = \frac{\partial}{\partial\Delta_a}\Big(\sum_{i=1}^n r_i(\Delta)\Big)\Big|_{\Delta=0} = \frac{\partial}{\partial\Delta_a}(1) = 0, \tag{37}$$

since softmax outputs sum to one for all $\Delta$.

(ii) Differentiate the gradient once more:

$$\frac{\partial^2\mathcal{K}}{\partial\Delta_a\,\partial\Delta_b} = \sum_{i=1}^n \frac{\partial^2 r_i}{\partial\Delta_a\,\partial\Delta_b}\left(\log\frac{r_i}{p_i} + 1\right) + \sum_{i=1}^n \frac{\partial r_i}{\partial\Delta_a}\frac{1}{r_i}\frac{\partial r_i}{\partial\Delta_b}. \tag{38}$$

At $\Delta = 0$, the first sum becomes $\sum_i \partial^2 r_i / \partial \Delta_a \partial \Delta_b$ (since $\log(r_i/p_i) = 0$), which is zero because $\sum_i r_i(\Delta) \equiv 1$ for all $\Delta$. Thus,

$$\left[ \nabla^2 \mathcal{K}(0) \right]_{ab} = \sum_{i=1}^{n} \frac{1}{p_i} \left[ \frac{\partial r_i}{\partial \Delta_a} \right]_{\Delta=0} \left[ \frac{\partial r_i}{\partial \Delta_b} \right]_{\Delta=0}. \tag{39}$$

It remains to compute the Jacobian of softmax at $v_{t-1}$:

$$J_{ia} := \left[ \frac{\partial r_i}{\partial \Delta_a} \right]_{\Delta=0} = \frac{\partial}{\partial v_a} \left( \frac{e^{v_i}}{\sum_j e^{v_j}} \right) \Big|_{v=v_{t-1}} = p_i \left( \mathbf{1}\{i = a\} - p_a \right). \tag{40}$$

Therefore,

$$\left[ \nabla^2 \mathcal{K}(0) \right]_{ab} = \sum_{i=1}^{n} \frac{1}{p_i} J_{ia} J_{ib} = \sum_{i=1}^{n} p_i \left( \mathbf{1}\{i = a\} - p_a \right)\left( \mathbf{1}\{i = b\} - p_b \right). \tag{41}$$

Expanding the sum gives

$$\sum_i p_i \mathbf{1}\{i = a\} \mathbf{1}\{i = b\} - p_b \sum_i p_i \mathbf{1}\{i = a\} - p_a \sum_i p_i \mathbf{1}\{i = b\} + p_a p_b \sum_i p_i. \tag{42}$$

Since $\sum_i p_i = 1$ and $\sum_i p_i \mathbf{1}\{i = a\} = p_a$, this equals

$$\delta_{ab}\, p_a \; - \; p_a p_b \; - \; p_a p_b \; + \; p_a p_b \; = \; \delta_{ab}\, p_a \; - \; p_a p_b, \tag{43}$$

i.e.

$$\nabla^2 \mathcal{K}(0) \; = \; \mathrm{diag}(p) \; - \; pp^\top. \tag{44}$$

(iii) By Taylor's theorem,

$$\mathcal{K}(\Delta) \; = \; \tfrac{1}{2} \Delta^\top \left( \mathrm{diag}(p) - pp^\top \right) \Delta \; + \; o(\|\Delta\|^2). \tag{45}$$

Finally, note the algebraic identity (weighted variance):

$$\Delta^\top \left( \mathrm{diag}(p) - pp^\top \right) \Delta = \sum_{i=1}^{n} p_i \Delta_i^2 - \left( \sum_{i=1}^{n} p_i \Delta_i \right)^2 = \sum_{i=1}^{n} p_i \left( \Delta_i - \mu \right)^2, \quad \mu := \sum_{i=1}^{n} p_i \Delta_i. \tag{46}$$

$$\square$$

Now we prove Proposition 3. Since the blockwise softmax-KL regularizer acts independently on each group $g$,

$$\mathcal{K}_{\mathrm{blk}}(\Delta) = \sum_{g=1}^{G} D_{\mathrm{KL}}\big( \mathrm{softmax}(v_{t-1}^{(g)} + \Delta^{(g)}) \,\|\, \mathrm{softmax}(v_{t-1}^{(g)}) \big), \tag{47}$$

where $\Delta^{(g)} = v_t^{(g)} - v_{t-1}^{(g)}$. Applying Proposition 5 to each group yields block Hessians

$$H^{(g)} = \mathrm{diag}(p^{(g)}) - p^{(g)}(p^{(g)})^\top, \tag{48}$$

which assemble into the block-diagonal

$$H = \mathrm{blkdiag}(H^{(1)}, \ldots, H^{(G)}). \tag{49}$$

The variance identity holds within each group.

## C EXPERIMENTS

### C.1 EXPERIMENTAL SETUP

**Datasets.** We use the real-world temporal Amazon Review dataset, which contains user reviews (treated as implicit interactions) on Amazon products over time.[2] We focus on three categories:

---

[2]https://amazon-reviews-2023.github.io/

Table 5: Dataset statistics.

|  |  | Total Users | New Users | Total Items | New Items | Total Interactions | Avg Seq Len | Sparsity |
|---|---|---|---|---|---|---|---|---|
| Instruments | $\mathcal{D}_1$ | 17,046 | 17,046 | 40,471 | 40,471 | 141,788 | 8.32 | 0.9998 |
|  | $\mathcal{D}_2$ | 1,772 | 1,183 | 8,346 | 2,900 | 13,197 | 7.45 | 0.9991 |
|  | $\mathcal{D}_3$ | 1,821 | 1,265 | 8,325 | 2,909 | 13,334 | 7.32 | 0.9991 |
|  | $\mathcal{D}_4$ | 2,289 | 1,684 | 9,617 | 3,864 | 18,811 | 8.22 | 0.9991 |
|  | $\mathcal{D}_5$ | 2,238 | 1,699 | 9,131 | 3,365 | 17,573 | 7.85 | 0.9991 |
|  | $\mathcal{D}_{1:5}$ | 22,877 | NA | 53,509 | NA | 204,703 | NA | NA |
| Movies & TVs | $\mathcal{D}_1$ | 17,928 | 17,928 | 39,228 | 39,228 | 190,411 | 10.62 | 0.9997 |
|  | $\mathcal{D}_2$ | 1,866 | 1,141 | 11,612 | 1,479 | 17,665 | 9.47 | 0.9992 |
|  | $\mathcal{D}_3$ | 2,106 | 1,200 | 12,658 | 1,926 | 19,874 | 9.44 | 0.9993 |
|  | $\mathcal{D}_4$ | 2,284 | 1,357 | 13,788 | 1,882 | 22,929 | 10.04 | 0.9993 |
|  | $\mathcal{D}_5$ | 2,332 | 1,552 | 13,491 | 1,559 | 22,225 | 9.53 | 0.9993 |
|  | $\mathcal{D}_{1:5}$ | 23,178 | NA | 46,074 | NA | 273,104 | NA | NA |
| Books | $\mathcal{D}_1$ | 15,406 | 15,406 | 35,984 | 35,984 | 164,858 | 10.7 | 0.9997 |
|  | $\mathcal{D}_2$ | 1,807 | 618 | 7,155 | 2,711 | 13,918 | 7.7 | 0.9989 |
|  | $\mathcal{D}_3$ | 1,672 | 619 | 6,484 | 2,278 | 12,395 | 7.41 | 0.9989 |
|  | $\mathcal{D}_4$ | 1,948 | 650 | 7,154 | 2,657 | 14,824 | 7.61 | 0.9989 |
|  | $\mathcal{D}_5$ | 1,652 | 1,025 | 5,913 | 2,274 | 11,990 | 7.26 | 0.9988 |
|  | $\mathcal{D}_{1:5}$ | 18,318 | NA | 45,904 | NA | 217,985 | NA | NA |

Musical Instruments, Movies & TV, and Books. For Instruments and Movies & TV, we use data from 2019–2023; for Books, we use 2022–2023. We take 60% of the data as pretraining $\mathcal{D}_1$ and split the remaining 40% into four equal incremental stages, $\mathcal{D}_2, ..., \mathcal{D}_5$. For each incremental stage, we filter out users with fewer than five interactions. This ensures leave-one-out evaluation is feasible and makes incremental data even smaller than pretraining data, simulating real-world scenarios. Table 5 summarizes dataset statistics, including the number of users, items, and interactions at each stage, average sequence length, and sparsity.

**Evaluation.** For each $\mathcal{D}_t$, we apply leave-one-out per user: the second-to-last item is used for validation and the last item is reserved for testing. Following prior work (Wang et al., 2024; Bao et al., 2025a), we construct multiple training pairs $(x_u, y_u)$ per user $u$ using a sliding window of size 20. The LLM trained on $\mathcal{D}_1$ serves as the pretrained model for all compared methods. At each stage $t = 2, \ldots, 5$, after fine-tuning, the LLM autoregressively generates 10 items given the user history in the test pair. Generation uses constrained beam search restricted to valid item tokens, making it efficient and widely adopted in prior work (Wang et al., 2024; Rajput et al., 2023). With these 10 items, we evaluate against the ground-truth item and report Hit@5, Hit@10, NDCG@5, and NDCG@10, averaged over $\mathcal{D}_2, \ldots, \mathcal{D}_4$.

**Metrics.** *Hit@k* measures whether the ground-truth item appears among the $k$ generated items. For a user $u$ with ground-truth item $y_u$ and a ranked list of predictions $R_u$,

$$\text{Hit@}k(u) = \begin{cases} 1 & \text{if } y_u \in R_u[1:k], \\ 0 & \text{otherwise.} \end{cases}$$

*NDCG@k* (Normalized Discounted Cumulative Gain) additionally accounts for the position of the ground-truth item, giving higher credit when it appears closer to the top:

$$\text{NDCG@}k(u) = \begin{cases} \frac{1}{\log_2(\text{rank}(y_u)+1)} & \text{if } y_u \in R_u[1:k], \\ 0 & \text{otherwise,} \end{cases}$$

Hit@$k$ captures whether the correct item is recommended at all, while NDCG@$k$ rewards ranking it higher in the list. We report averages of Hit@$k$ and NDCG@$k$ across all users, with $k \in 5, 10$.

## C.2 LEARNING RATE FOR CONTINUAL STAGES

Incremental blocks are much smaller than the pretraining set $\mathcal{D}_1$ (see Appendix C.1), making performance sensitive to the learning rate. Figure 4 reports results for Single Evolving LoRA under different learning rates on incremental blocks. Using the pretraining rate (0.0002; `lr*`=1.0) performs worse than not updating on new data, likely due to overfitting. The best performance is achieved with `lr*`=0.05 or `lr*`=0.1, which aligns with the relative block size $|\mathcal{D}_t|/|\mathcal{D}_1| \approx 0.1$. This suggests that learning rates for incremental blocks should be scaled with respect to data size.

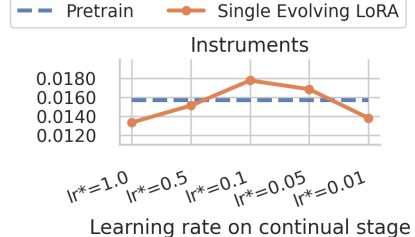

Figure 4: Impact of the learning rate for continual data on model performance.

Table 6: Comparison of LLM-based and traditional methods in continual recommendation.

|  |  | Instruments | Movies & TVs | Books |
|---|---|---|---|---|
| Traditional two-tower | Pretrain | 0.0153 | 0.0028 | 0.0041 |
|  | Fine-tuning | 0.0180 | 0.0114 | 0.0218 |
|  | Contrastive | 0.0177 | 0.0101 | 0.0272 |
|  | Contrastive + PIW | 0.0193 | 0.0113 | 0.0243 |
|  | PISA | 0.0194 | 0.0106 | 0.0301 |
| LLM-based | Pretrain | 0.0157 | 0.0160 | 0.0235 |
|  | Fine-tuning (w/ LoRA) | 0.0178 | 0.0169 | 0.0414 |
|  | PESO | 0.0190 | 0.0173 | 0.0422 |

### C.3 COMPARISON WITH TRADITIONAL CONTINUAL RECOMMENDER SYSTEMS

We compare our LLM-based methods (pretrain, single evolving LoRA, and PESO) against two-tower methods with LightGCN (He et al., 2020) as backbone, including Pretrain, Fine-tuning, Contrastive (Wang et al., 2021), Contrastive+PIW (Wang et al., 2023b), and PISA (Yoo et al., 2025a). Two-tower models use explicit user and item embeddings, and their continual methods mitigate forgetting by regularizing user embeddings against past versions: Contrastive maximizes mutual information between past and current embeddings, Contrastive+PIW further adapts the regularization weights per user, and PISA combines stability and plasticity regularization.

Table 6 reports results averaged across time stages and metrics. First, LLM-based recommenders (both pretrain and continual) generally outperform traditional methods, highlighting the generalization ability and knowledge transfer benefits of LLMs. On Instruments, however, the performance gap is smaller, suggesting that explicit dual modeling of users and items still provides benefits for capturing collaborative signals. It is worth noting that there also remains considerable headroom for LLM-based models if larger beam sizes are used during generation.

Second, While PESO outperforms traditional continual methods in absolute terms, the relative gains of continual techniques over their respective pretraining baselines are larger in traditional settings. This is likely because two-tower methods explicitly capture preference shifts through user embeddings, supporting our view that modeling user preference drift is crucial in continual recommendation. At the same time, it underscores the difficulty of capturing such dynamics in LLM-based methods, pointing to an important direction for future research.

### C.4 QUANTIFICATION OF DISTRIBUTION SHIFT

To validate the realism of our continual learning formulation, we explicitly quantified the user preference drift between data blocks on the Instruments dataset. We employed a domain discrimination approach:

1. We embed user interaction sequences into fixed-dimensional vectors using pretrained codebooks.

2. For each pair of blocks $(t-1)$ and $(t)$, we train a binary logistic regression classifier to distinguish samples from the two blocks.

3. We compute the **Drift Score** $\delta(t-1,t) = 2(\text{AUC} - 0.5) \in [0,1]$, where 0 implies identical distributions and 1 implies completely separable distributions.

Table 7 reports both the step-wise drift $\delta(t-1,t)$ and the cumulative drift from the base block $\delta(0,t)$. The results show non-trivial step-wise drift and, crucially, a steady increase in cumulative drift (reaching 0.457 at $t = 4$). This confirms that user preferences structurally evolve away from the initial state, validating our experimental setup.

Table 7: Quantification of distribution shift (Drift Score) on the Instruments dataset.

| Measure | $t=1$ | $t=2$ | $t=3$ | $t=4$ |
|---|---|---|---|---|
| Step-wise $\delta(t-1,t)$ | 0.200 | 0.060 | 0.240 | 0.090 |
| Cumulative $\delta(0,t)$ | 0.200 | 0.311 | 0.342 | 0.457 |

## C.5 Comparison with Additional Baselines

### C.5.1 Comparison with Standard Training Paradigms (Full-Parameter Fine-Tuning & Retraining)

To validate the effectiveness of our LoRA-based sequential fine-tuning approach, we compared it against two traditional training paradigms:

1. **Full-Parameter Fine-Tuning:** Updating *all* LLM parameters sequentially on new data blocks.

2. **Retraining (with LoRA):** At each stage $t$, we restart from the same pretrained LLM (i.e., without warm-starting from stage $t-1$) and fine-tune it on the *cumulative dataset* ($\mathcal{D}_1 \cup \cdots \cup \mathcal{D}_t$).

As shown in Table 8, **Single Evolving LoRA** consistently outperforms both approaches.

- **vs. Full-Parameter Fine-Tuning:** Full-parameter updates suffer from a dilemma: high learning rates ($2e-5$) lead to catastrophic forgetting, while low rates ($2e-6$) result in insufficient adaptation. LoRA acts as a structural regularizer, mitigating forgetting while enabling effective adaptation.

- **vs. Retraining:** While full retraining outperforms static pretraining, it underperforms sequential fine-tuning. This aligns with prior work (Yoo et al., 2025a), suggesting that sequential updates naturally prioritize recent preference signals, whereas full retraining treats old and new data equally, diluting the signal of evolving interests.

Table 8: Comparison with Standard Training Paradigms (Full Fine-Tuning and Full Retraining).

| Method | Hit@5 | Hit@10 | NDCG@5 | NDCG@10 |
|---|---|---|---|---|
| Pretrain (Static) | 0.0166 | 0.0216 | 0.0115 | 0.0131 |
| Full Retraining (Cumulative Data) | 0.0170 | 0.0231 | 0.0121 | 0.0141 |
| Full Fine-Tuning ($lr = 2e-5$) | 0.0142 | 0.0228 | 0.0099 | 0.0127 |
| Full Fine-Tuning ($lr = 2e-6$) | 0.0171 | 0.0254 | 0.0122 | 0.0149 |
| **Single Evolving LoRA** (LoRA Fine-Tuning) | **0.0181** | **0.0253** | **0.0127** | **0.0150** |

### C.5.2 Comparison with Additional Continual LoRA Methods

We compared PESO against O-LoRA (Wang et al., 2023a), AM-LoRA (Liu et al., 2024a), and LSAT (Shi et al., 2024) on the Instrument dataset. O-LoRA and AM-LoRA belong to the cumulative family but use *orthogonality* or *attention mechanisms* to combine adapters, while LSAT utilizes *adapter interpolation*. As shown in Table 9, PESO consistently outperforms all of them. This supports our claim that explicitly maintaining discrete adapters is less effective for gradual preference drift than our proximal regularization approach.

Table 9: Comparison with recent Continual PEFT methods on Instruments.

| Method | Hit@5 | Hit@10 | NDCG@5 | NDCG@10 |
|---|---|---|---|---|
| Single Evolving LoRA | 0.0181 | 0.0253 | 0.0127 | 0.0150 |
| Cumulative LoRA | 0.0182 | 0.0260 | 0.0129 | 0.0154 |
| O-LoRA | 0.0191 | 0.0259 | 0.0134 | 0.0156 |
| AM-LoRA | 0.0182 | 0.0240 | 0.0125 | 0.0144 |
| LSAT | 0.0164 | 0.0250 | 0.0117 | 0.0144 |
| LSAT (+ Param Inheritance) | 0.0183 | 0.0254 | 0.0130 | 0.0153 |
| **PESO** | **0.0193** | **0.0268** | **0.0138** | **0.0162** |

## C.6 PERFORMANCE ON LC-REC BACKBONE

To demonstrate robustness across architectures, we evaluated PESO using the LC-REC backbone (Zheng et al., 2024). As shown in Table 10, PESO maintains its superiority over baselines.

Table 10: Performance comparison using the LC-REC backbone.

| Method | Hit@5 | Hit@10 | NDCG@5 | NDCG@10 |
|---|---|---|---|---|
| Single Evolving LoRA | 0.0164 | 0.0249 | 0.0119 | 0.0146 |
| Cumulative LoRA | 0.0178 | 0.0249 | 0.0122 | 0.0145 |
| SD-LoRA | 0.0185 | 0.0256 | 0.0127 | 0.0150 |
| **PESO** | **0.0179** | **0.0266** | **0.0130** | **0.0158** |

## C.7 PERFORMANCE ON NON-E-COMMERCE DATASET (YELP)

To further explore non-e-commerce domains, we evaluated PESO on the Yelp dataset, where interactions correspond to user check-ins at locations, using the same data-splitting strategy as in our main experiments. As shown in Table 11, PESO consistently outperforms strong competitors, including Single Evolving LoRA and SD-LoRA.

This result is particularly notable given that, unlike Amazon products which feature detailed textual descriptions, Yelp locations often lack deep semantic content (consisting primarily of names like "Pizza Hut" or coarse categories like "Pizza" or "Restaurant"). This demonstrates the robustness of our method, showing it remains highly effective even in settings with limited semantic richness.

Table 11: Performance comparison on the Yelp dataset.

| Methods | Hit@5 | Hit@10 | NDCG@5 | NDCG@10 |
|---|---|---|---|---|
| Pretrain | 0.0201 | 0.0309 | 0.0126 | 0.0161 |
| Single Evolving LoRA | 0.0290 | 0.0442 | 0.0190 | 0.0239 |
| SD-LoRA | 0.0279 | 0.0432 | 0.0168 | 0.0230 |
| **PESO** | **0.0302** | **0.0454** | **0.0199** | **0.0248** |

## C.8 EXPLICIT MEASUREMENT OF FORGETTING

We measured the performance drop on past blocks to analyze forgetting behavior. Table 12 shows the difference between the final model's performance on $D_t$ and its initial performance at time $t$. While PESO shows selective forgetting on intermediate blocks (allowing it to shed obsolete trends), it achieves the highest overall performance and best retrieval for dormant users, indicating that this forgetting is benign and adaptive rather than catastrophic.

Table 12: Performance drop on past blocks (lower is strictly less forgetting, but may imply rigidity).

| Method | Drop on $D_0$ | Drop on $D_1$ | Drop on $D_2$ | Drop on $D_3$ |
|---|---|---|---|---|
| Single Evolving LoRA | 0.0062 | 0.0087 | 0.0042 | 0.0031 |
| Cumulative LoRA | 0.0060 | 0.0062 | 0.0035 | 0.0060 |
| **PESO** | 0.0062 | 0.0107 | 0.0048 | 0.0045 |

## D  EFFICIENCY ANALYSIS

PESO introduces negligible overhead compared to baselines:

- **Storage Complexity:** PESO stores only one previous LoRA adapter, resulting in $O(1)$ storage complexity relative to the number of stages. In contrast, Cumulative LoRA grows linearly $O(T)$ as it must store all past adapters.
- **Computational Complexity:** PESO adds only a lightweight quadratic/KL penalty to the loss. This requires no additional forward passes. In practice, we observed no measurable slowdown in training time compared to standard Single LoRA fine-tuning.

## E  DISCUSSION ON PROMPT TUNING VS. LoRA

Prompt-tuning–based PEFT methods typically learn a prompt pool and dynamically retrieve the most relevant prompts for each input, inserting them into the input or intermediate representations without updating backbone weights (Wang et al., 2022) This introduces inference overhead because the model must compute query features and perform similarity matching over a growing prompt pool at inference time. The inference-inefficiency is even more severe in our generative recommendation setting: autoregressive generation requires many forward passes per prediction, and each step would need repeated prompt retrieval. Recent studies in vision (Wu et al., 2025; Liang & Li, 2024) also report that LoRA-based methods generally outperform prompt-based approaches in large-scale tasks, making LoRA the preferred PEFT technique.

## F  ADDITIONAL RELATED WORK

In modern machine learning research—including recommender systems, personalized LLM agents, and agentic memory (Wei et al., 2025b; 2026; Bei et al., 2026; Ning et al.; Liu et al., 2025b;c;d; 2024b; Li et al., 2025b;c; Lin et al., 2024a; Qiu et al., 2025a;b;c; He et al., 2026; Lee et al., 2025a; Xu et al., 2024)—temporal adaptation under distribution shift (Zeng et al., 2026b) has been extensively studied. Our work focuses on parameter-efficient tuning and regularization/distillation to mitigate forgetting and lock-in in continual recommendation settings. It complements this line of research by maintaining a single evolving LoRA adapter and applying a proximal regularizer that provides data-aware control of the stability–plasticity trade-off.

## G  PROMPT

We show below the template used in all experiments. Notably, `<a_[i1]><b_[j1]><c_[k1]><d_[l1]>` represents one user–item interaction encoded as four semantic-ID tokens. For instance, `<a_144><b_72><c_103><d_217>` is one such tuple describing a single interacted item (Rajput et al., 2023; Wang et al., 2024).

```
Below is an instruction that describes a task.
Write a response that appropriately completes the request.\n\n

### Instruction:\n
Based on the items that the user has interacted with:
<a_[i1]><b_[j1]><c_[k1]><d_[l1]>,
```

```
<a_[i2]><b_[j2]><c_[k2]><d_[l2]>,
...,
<a_[iN]><b_[jN]><c_[kN]><d_[lN]>,
can you determine what item would be recommended to the user next?\n\n
### Response:
```

## H    USE OF LARGE LANGUAGE MODELS

LLMs were used only for writing polish (grammar and clarity). All content was reviewed and approved by the authors. LLMs did not contribute to research ideation, algorithm design, implementation, or analysis.

