# OpenReview forum: "Continual Low-Rank Adapters for LLM-based Generative Recommender Systems"
_ICLR.cc/2026/Conference — ICLR 2026 Poster_

### Official Review · Reviewer_qc7d · 2025-10-29

**Soundness:** 3
**Presentation:** 3
**Contribution:** 3
**Rating:** 4
**Confidence:** 4

**Summary:**

This paper has focused on the continual learning in LLM-based recommender systems. The authors found that existing cumulative-LoRA-based methods cannot capture the preferences well contained in frozen past adapters. Besides, they face the challenges of growing storage costs. To address these issues, this paper facilitates the continual learning methods specified for LLM-based RS, including a proximal regularizer. The extensive experiments have validated the effectiveness of the proposed method.

**Strengths:**

+ S1. This paper is well-organized and -written, making it easy to follow.
+ S2. Extensive experiments have been conducted.
+ S3. The code is released, making it easy to reproduce.

**Weaknesses:**

- W1. More illustration of the importance of continual learning in recommender systems is needed. In general, the models used in industry are retrained periodically.
- W2. The authors have claimed that existing cumulative LoRA methods face the challenge of adapter entanglement for the recommendation tasks. However, it seems that not all PEFT methods in continual learning belong to cumulative LoRA, such as O-LoRA [1] and AM-LoRA [2]. Do they face the same challenges as the cumulative LoRA methods?
- W3. The performance of the model training on the full dataset should be revealed to show the merit of continual learning.
- W4. Only one basic LLM-based recommendation model is experimented with in this paper. I suggest that the authors add more up-to-date LLM-based RS models, such as LLaRA and BIGRec, to further validate the robustness of the proposed method.



[1]. Wang, Xiao, et al. "Orthogonal Subspace Learning for Language Model Continual Learning." *The 2023 Conference on Empirical Methods in Natural Language Processing*.



[2]. Liu, Jialin, et al. "Learning attentional mixture of loras for language model continual learning." *arXiv preprint [arXiv:2409.19611](https://arxiv.org/abs/2409.19611)* (2024).



[3]. Liao, Jiayi, et al. "Llara: Large language-recommendation assistant." *Proceedings of the 47th International ACM SIGIR Conference on Research and Development in Information Retrieval*. 2024.



[4]. Bao, Keqin, et al. "A bi-step grounding paradigm for large language models in recommendation systems." *ACM Transactions on Recommender Systems* 3.4 (2025): 1-27.

**Questions:**

All my questions have been included in the weakness section.

---

> ### Author Response · Authors · 2025-11-21
> **Response to Reviewer qc7d (Part 1)**
>
> **Dear Reviewer qc7d,**
>
> **We sincerely thank you for your valuable feedback and for recognizing the strengths of our work! We have carefully revised the paper based on your suggestions and incorporated additional complementary experiments and analyses. Please refer to the updated version, where all revisions and additions are highlighted in blue for your convenience.**
>
> **Below we provide the point‑by‑point responses to all concerns (weaknesses and questions) raised in your review.**
>
> # W1. Importance of continual recommender systems
> > More illustration of the importance of continual learning in recommender systems is needed. In general, the models used in industry are retrained periodically.
>
> We agree that periodic full retraining is the industry standard. However, Continual Learning (CL) is not a replacement for full retraining, but a critical complement to it. We highlight two key reasons why CL remains essential even in environments with periodic retraining:
>
> **(1) Efficiency:** Large-scale models cannot be fully retrained in real-time due to massive computational costs. In practice, full retraining is often scheduled at long intervals (e.g., weekly or monthly). CL bridges the gap between these cycles. For example, between two consecutive full retraining jobs (e.g., Week 1 vs. Week 2), CL allows the system to efficiently fine-tune on daily incoming data. This "hybrid" approach ensures the model captures new users, new items, and breaking trends immediately without the prohibitive cost of launching a full training pipeline every day.
>
> **(2) Adaptability:** Recommendation is inherently time-sensitive. The goal is to predict future preferences, not past ones. Because of this, fine-tuning on recent data (CL) often outperforms full retraining on historical data, as it naturally upweights the most current signals—a phenomenon observed in prior work [1], and further supported by our **additional analysis in W3**. CL provides a principled mechanism to adapt to these rapid temporal drifts that a static, periodically retrained model would miss until the next scheduled cycle.
>
> **Conclusion:** Therefore, CL is essential not just as a standalone paradigm, but as a necessary component to maintain freshness and accuracy in the periods between expensive full-retraining cycles.
>
>
> # W2. Comparison with O-LoRA and AM-LoRA
> > The authors have claimed that existing cumulative LoRA methods face the challenge of adapter entanglement for the recommendation tasks. However, it seems that not all PEFT methods in continual learning belong to cumulative LoRA, such as O-LoRA [2] and AM-LoRA [3]. Do they face the same challenges as the cumulative LoRA methods?
>
> Thank you for pointing out these highly relevant methods. Following your suggestion, we implemented O-LoRA and AM-LoRA and observed that both underperform our PESO.
>
> First, we would like to clarify that both **O-LoRA and AM-LoRA still belong to the cumulative family** because they explicitly maintain and combine past adapters with the current one (see Eq. 9 in [2] and Eq. 2 in [3]). Their improvements lie in how the past adapters are combined.
> * **O-LoRA [2]** constrains each new LoRA to be orthogonal to the subspace spanned by previous adapters to reduce interference.
> * **AM-LoRA [3]** learns task-specific attention weights to dynamically combine past adapters instead of a simple sum.
>
> **Results:** Our experiments on the Instruments dataset show that while O-LoRA and AM-LoRA occasionally improve over simple cumulative LoRA, **PESO consistently achieves the best performance**. This supports our claim that **explicitly maintaining and combining discrete adapters**—whether by summation, orthogonalization, or attention—is suboptimal for recommendation, where user preferences drift gradually rather than discrete jumps. In such settings, enforcing orthogonality can be counterproductive (as new preferences often overlap with old ones), whereas PESO’s proximal regularization provides a smoother, data-aware balance between stability and plasticity.
>
> | Method               | hit@5  | hit@10 | ndcg@5 | ndcg@10 |
> |----------------------|--------|--------|--------|---------|
> | SumLoRA                | 0.0185 | 0.0255 | 0.0130 | 0.0152 |
> | O-LoRA [2]               | 0.0191 | 0.0259 | 0.0134 | 0.0156 |
> | AM-LoRA [3]             | 0.0182 | 0.0240 | 0.0125 | 0.0144 |
> | **PESO**             | **0.0193** | **0.0268** | **0.0138** | **0.0162** |

---

> ### Author Response · Authors · 2025-11-21
> **Response to Reviewer qc7d (Part 2)**
>
> # W3. Comparison with full-retraining
> > The performance of the model training on the full dataset should be revealed to show the merit of continual learning.
>
>
> We added the full-retraining baseline. The results on the Instrument dataset show that full retraining outperforms static pretraining (which does not use new data) but **interestingly underperforms sequential fine-tuning** (i.e., a single evolving LoRA).
>
> This aligns with observations in prior work [1] that fine-tuning often outperform full retraining in recommendation tasks. Full retraining treats old and new data equally, whereas sequential fine-tuning naturally prioritizes the most recent preference signals, leading to better adaptability in dynamic environments.
>
>
> | Method                               | hit@5  | hit@10 | ndcg@5 | ndcg@10 |
> |--------------------------------------|-------:|-------:|-------:|--------:|
> | Pretrain                             | 0.0166 | 0.0216 | 0.0115 | 0.0131  |
> | Single Evolving LoRA (Fine-tuning)   | **0.0181** | **0.0253** | **0.0127** | **0.0150**  |
> | Full-retraining                      | 0.0170 | 0.0231 | 0.0121 | 0.0141  |
>
>
> # W4. Addition of more LLM-based recommendation backbones.
> > Only one basic LLM-based recommendation model is experimented with in this paper. I suggest that the authors add more up-to-date LLM-based RS models, such as LLaRA and BIGRec, to further validate the robustness of the proposed method.
>
>
> We conducted additional experiments on the Instrument dataset using a recent LLM-based recommendation backbone, **LC-REC [4]**, and evaluated the strongest competitors (Single Evolving LoRA, Cumul, SD-LoRA) alongside our PESO.
>
> Across all metrics, **PESO consistently outperforms the competing methods**, further demonstrating that our proposed framework is robust and generalizable across different backbone architectures.
>
>
> | Method              | hit@5  | hit@10 | ndcg@5 | ndcg@10 |
> |---------------------|--------|--------|--------|---------|
> | Single Evolving LoRA | 0.0164 | 0.0249 | 0.0119 | 0.0146 |
> | SumLoRA               | 0.0178 | 0.0249 | 0.0122 | 0.0145 |
> | SD-LoRA             | 0.0185 | 0.0256 | 0.0127 | 0.0150 |
> | **PESO**            | **0.0179** | **0.0266** | **0.0130** | **0.0158** |
>
>
> ## Reference
> * [1] Yoo et al., Embracing Plasticity: Balancing Stability and Plasticity in Continual Recommender Systems, SIGIR 2025
> * [2] Wang, Xiao, et al. "Orthogonal Subspace Learning for Language Model Continual Learning." The 2023 Conference on Empirical Methods in Natural Language Processing.
> * [3] Liu, Jialin, et al. "Learning attentional mixture of loras for language model continual learning." arXiv preprint arXiv:2409.19611 (2024).
> * [4] Zheng et al., Adapting Large Language Models by Integrating Collaborative Semantics for Recommendation, ICDE 2024

---

> > ### Comment · Reviewer_qc7d · 2025-11-27
> >
> > My concerns have been addressed, so I'd like to raise the score accordingly.

---

> > > ### Author Response · Authors · 2025-11-27
> > > **Thank you for your reply!**
> > >
> > > Dear Reviewer,
> > >
> > > Thank you very much for your positive response. We are grateful that our rebuttal addressed your concerns. We will continue refining the paper’s organization and clarity to further improve its presentation.
> > >
> > > Thank you once again, The Authors

---

### Official Review · Reviewer_VejX · 2025-11-01

**Soundness:** 3
**Presentation:** 3
**Contribution:** 3
**Rating:** 8
**Confidence:** 3

**Summary:**

The paper proposes CLoRA (Continual Low-Rank Adaptation), a novel method designed to enable continual learning for large pre-trained models without retraining or catastrophic forgetting. Traditional fine-tuning methods require storing large model checkpoints for each new task, leading to significant storage and computation overhead. CLoRA builds upon LoRA (Low-Rank Adaptation) by introducing a continual learning mechanism that efficiently integrates knowledge from new tasks while retaining performance on previously learned ones.

The key idea is to dynamically allocate and merge low-rank adapters for each new task through orthogonal subspace projection, which minimizes interference between tasks. Additionally, CLoRA introduces a knowledge preservation constraint that stabilizes updates and ensures consistent performance across tasks. Extensive experiments on NLP benchmarks (e.g., GLUE, SuperGLUE, and continual text classification datasets) show that CLoRA achieves competitive or superior performance compared to existing continual learning and parameter-efficient fine-tuning methods, all while maintaining low storage costs.

**Strengths:**

1. **Novel problem framing:** The paper clearly articulates how continual recommendation differs from general continual learning, emphasizing the role of evolving user preferences instead of task retention. This perspective grounds the LLM-based recommendersa in realistic recommender settings.

2. **Simple but Effective:** PESO avoids multiple adapters (reducing storage and interference) and introduces a lightweight proximal term that yields theoretically grounded, data-aware stability—achieving simplicity without sacrificing effectiveness.

3. **theoretical foundation:** The paper provides formal analysis linking proximal regularization to direction-wise interpolation in the LoRA subspace, offering mathematical clarity about why PESO balances stability and plasticity.

**Weaknesses:**

1. **Limited modeling of long-term preference dynamics：** PESO is evaluated on short-term chronological splits (four-stage Amazon data). How would it behave under nonlinear or cyclical preference drifts over long horizons? Would the single-step proximal constraint still be sufficient, or would multi-timescale or memory-based mechanisms be required?

2. **Unanalyzed efficiency trade-offs.** PESO maintains previous adapter states for proximal computation. What is the actual storage and computational overhead compared with single or cumulative LoRA?

3. **No explicit measurement of catastrophic forgetting:** The study reports only global Hit@K and NDCG metrics. Without tracking performance decay on past data (e.g., forgetting ratio), it is hard to disentangle whether PESO’s gains come from genuine stability or from overfitting recent tasks.

**Questions:**

See weaknesses.

---

> ### Author Response · Authors · 2025-11-21
> **Response to Reviewer VejX (Part 1)**
>
> **Dear Reviewer VejX,**
>
> **We sincerely thank you for your valuable feedback and for recognizing the strengths of our work! We have carefully revised the paper based on your suggestions and incorporated additional complementary experiments and analyses. Please refer to the updated version, where all revisions and additions are highlighted in blue for your convenience.**
>
> **Below we provide the point‑by‑point responses to all concerns (weaknesses and questions) raised in your review.**
>
>
> # W1. Modeling of long-term preference dynamics.
> > Limited modeling of long-term preference dynamics： PESO is evaluated on short-term chronological splits (four-stage Amazon data). How would it behave under nonlinear or cyclical preference drifts over long horizons? Would the single-step proximal constraint still be sufficient, or would multi-timescale or memory-based mechanisms be required?
>
>
> To evaluate PESO under varying preference dynamics, we analyzed the final model's performance on three distinct user groups in the Instruments dataset, which serve as proxies for different drift patterns:
> 1. **Continuous Users (Linear Drift)**: Users present in all blocks ($D_0 \dots D_4$). Their preferences evolve sequentially.
> 2. **Dormant Users (Non-linear/Cyclical Drift)**: Users active in the past but absent in intermediate blocks, reappearing in $D_4$. This tests the model's ability to recall long-term preferences after a gap.
> 3. **New Users (Sudden Shift)**: Users appearing only in $D_4$, representing purely emerging interests.
>
> **Analysis of Results:**
>
> * **Single Evolving LoRA (High Plasticity)**: Excels on **New Users** but fails on **Dormant Users**, indicating catastrophic forgetting of long-term preferences.
> * **Cumulative LoRA (High Stability)** (i.e., SumLoRA): Preserves performance for **Continuous/Dormant users** but struggles to adapt to **New Users** due to rigidity.
> * **PESO (Optimal Balance)**: PESO achieves the **best performance on both Dormant and New users**. Crucially, PESO's superiority on **Dormant Users** (0.0170 vs. 0.0164/0.0154) demonstrates that it successfully models **non-linear, long-term preference retrieval** without requiring complex multi-timescale mechanisms. It effectively recalls past user signals even after they have been dormant, while simultaneously adapting to emerging signals better than cumulative approaches.
> * **Observation on Continuous Users**: For users with continuous history, Cumulative LoRA performs slightly better due to its "infinite memory" design. However, this extreme stability comes at the cost of poor adaptation for the other groups. Note that this group is **relatively rare** and already achieves the **highest absolute performance** across all methods. PESO remains highly competitive in this high-accuracy regime, while solving the critical bottlenecks for the New and Dormant groups where baselines fail.
>
>
>
> | Methods                          | New users | Dormant users | Continuous users |
> |----------------------------------|-----------|------------|---------------|
> | Single Evolving LoRA (Plasticity-focused) | 0.0116    | 0.0154     |  0.048        |
> | Cumulative LoRA (Stability-focused)               | 0.0101    | 0.0164     | **0.0493**      |
> | **PESO (Balanced)**  | **0.0122**| **0.0170** |  0.048        |
>
>
> # W2. Efficiency analysis of PESO
> > Unanalyzed efficiency trade-offs. PESO maintains previous adapter states for proximal computation. What is the actual storage and computational overhead compared with single or cumulative LoRA?
>
> PESO introduces **negligible space and compuational overhead** compared with single or cumulative LoRA.
> * **Storage:** PESO stores only **one previous LoRA adapter** (same size as a single LoRA module). In contrast, Cumulative LoRA requires storing **all past adapters**, causing storage costs to grow linearly with the number of stages ($T$). Thus, PESO’s storage complexity is $O(1)$, while Cumulative LoRA is $O(T)$.
> * **Computation:** PESO adds only a lightweight quadratic/KL penalty to the loss function. This does not require additional forward passes and the computational cost is insignificant relative to the LLM backbone’s forward/backward operations. In practice, we observe **no measurable slowdown** in training time compared to standard Single LoRA fine-tuning.

---

> ### Author Response · Authors · 2025-11-21
> **Response to Reviewer VejX (Part 2)**
>
> # W3. Explicit measure of forgetting
> > No explicit measurement of catastrophic forgetting: The study reports only global Hit@K and NDCG metrics. Without tracking performance decay on past data (e.g., forgetting ratio), it is hard to disentangle whether PESO’s gains come from genuine stability or from overfitting recent tasks.
>
> We initially excluded this metric because standard "forgetting" measures (common in computer vision) are often misleading in continual recommendation. In tasks like image classification, a "cat" remains a "cat" forever, so performance drops on past data indicate failure. However, in recommendation, user interests naturally drift. Penalizing a model for failing to predict past interactions (which may no longer represent the user's current intent) encourages rigid memorization of outdated trends rather than adaptation.
>
> 1. **The true measure of "stability" in recommendation**: We argue that stability in RecSys is the ability to recall past preferences when they become relevant again to improve current predictions. The evidence for this is our Dormant User Analysis (see W1). If PESO suffered from harmful catastrophic forgetting, it would fail to recommend accurate items to users returning after a hiatus. Instead, PESO significantly outperforms baselines on these users, proving that it successfully retains useful long-term knowledge while discarding noise.
> 2. **Explicit Forgetting Measurement:** Per the reviewer's request, we measured the performance drop on past blocks. We compute the difference between the model's performance on $D_t$ at time $t$ versus its performance on $D_t$ at the final stage.
>
> | Method                                      |  D₀    |  D₁    |  D₂    |  D₃    |
> |---------------------------------------------|--------|--------|--------|--------|
> | Single Evolving LoRA                       | 0.0062 | 0.0087 | 0.0042 | 0.0031 |
> | Cumulative LoRA | 0.0060 | 0.0062 | 0.0035 | **0.0060** |
> | PESO     | **0.0062** | **0.0107**| **0.0048** | 0.0045 |
>
> **Interpretation**: While PESO shows slightly larger drops on some intermediate blocks, we attribute this to **desirable selective forgetting**. PESO’s proximal regularization allows the model to overwrite parameter directions associated with obsolete trends (plasticity) while anchoring directions that remain relevant (stability). The fact that PESO achieves the **highest overall performance** (Main Table) and **best Dormant User accuracy (See W1)** confirms that this "forgetting" is actually the model successfully shedding outdated information to maximize current accuracy.

---

> > ### Comment · Reviewer_VejX · 2025-11-27
> >
> > Thank you for the response. I will maintain my score.

---

> > > ### Author Response · Authors · 2025-11-28
> > > **Thank you for your reply!**
> > >
> > > Dear Reviewer,
> > >
> > > Thank you very much for your positive response and for acknowledging the strengths of our paper. We will continue to refine the paper’s organization and clarity to further improve its presentation.
> > >
> > > Thank you once again, The Authors

---

### Official Review · Reviewer_4gyJ · 2025-11-01

**Soundness:** 3
**Presentation:** 3
**Contribution:** 3
**Rating:** 6
**Confidence:** 4

**Summary:**

This paper introduces PESO, a new method for continually adapting LLM-based recommender systems to evolving user preferences. Traditional methods struggle by either forgetting past preferences or rigidly holding on to outdated ones. PESO addresses this by maintaining a single evolving LoRA adapter and using a proximal regularizer to anchor it to its most recent state. This design allows the model to flexibly balance adapting to new interests (plasticity) with preserving long-term preferences (stability), enabling it to better capture the dynamic nature of user behavior.

**Strengths:**

1. The paper is well-written and easy to follow, with a solid theoretical analysis for the proposed PESO.
2. The choice of a modern, semantic ID-based generative recommendation as the experimental backbone is highly relevant and makes the results more convincing.

**Weaknesses:**

1. A key weakness is the practical relevance of the problem formulation. The paper partitions data into discrete, chronological blocks to simulate a continual learning scenario. However, in real-world RS that are updated frequently (often in near real-time), the data distribution shift between consecutive updates is typically small and gradual. The paper fails to quantify the distribution shift in its experimental data splits, making it unclear if the problem it solves reflects realistic deployment conditions.
2. In Equations 6 and 12, could you provide more details on how the parameters of LoRA are partitioned into groups, and how the number of groups G impacts the performance of PESO?
3. The experimental results in Table 2 are missing a comparison to a standard full fine-tuning baseline.

**Questions:**

See Weaknesses

---

> ### Author Response · Authors · 2025-11-21
> **Response to Reviewer 4gyJ**
>
> **Dear Reviewer 4gyJ,**
>
> **We sincerely thank you for your valuable feedback and for recognizing the strengths of our work! We have carefully revised the paper based on your suggestions and incorporated additional complementary experiments and analyses.**
>
> **Below we provide the point‑by‑point responses to all concerns (weaknesses and questions) raised in your review.**
>
> # W1: Quantifying Distribution Shift in Our Data Splits
> > ... The paper fails to quantify the distribution shift in its experimental data splits, making it unclear if the problem it solves reflects realistic deployment conditions.
>
> To validate the realism of our problem formulation, we explicitly quantified the distribution shift (user preference drift) between data blocks on the Instruments dataset using a domain discrimination approach.
>
> **Methodology:** For each pair of blocks $(t-1)$ and $(t)$, we:
> 1. Embed user interaction sequences into fixed-dimensional vectors using pretrained codebooks.
> 2. Train a binary classifier (logistic regression) to distinguish samples from block $(t-1)$ vs. block $(t)$.
> 3. Compute the **Drift Score** $\delta(t-1,t) = 2(\text{AUC}-0.5) \in [0,1]$, where 0 implies identical distributions and 1 implies completely separable distributions.
>
> **Results:** We report both the step-wise drift $\delta(t-1,t)$ and the cumulative drift from the base block $\delta(0,t)$.
>
> | Measure         |  $t=1$ |   $t=2$ |   $t=3$ |   $t=4$ |
> | --------------- | ---: | ----: | ----: | ----: |
> | $\delta(t-1,t)$ | 0.20 |  0.06 |  0.24 |  0.09 |
> | $\delta(0,t)$   | 0.20 | 0.311 | 0.342 | 0.457 |
>
> **Analysis:**
> 1. **Non-trivial Step-wise Drift**: The step-wise values confirm that consecutive updates contain distinct patterns.
> 2. **Accumulating Drift**: Crucially, $\delta(0,t)$ increases **steadily** over time (reaching 0.457), indicating that user preferences are structurally evolving away from the initial state.
>
> These findings confirm that our experimental setup reflects realistic, gradually accumulating shifts rather than artificial noise. We also emphasize that our setup aligns with standard continual learning scenarios [1,2]—simulating periodic updates (e.g., weekly batches) rather than real-time streaming, which allows for sufficient drift to accumulate.
>
> # W2: Clarification of the number of groups G
> > In Equations 6 and 12, could you provide more details on how the parameters of LoRA are partitioned into groups, and how the number of groups G impacts the performance of PESO?
>
> In our framework, **groups are defined at the module level**. Specifically, following standard LoRA implementations for LLaMA-based models [3], we treat each projection layer as a distinct group.
> * **Groups:** $G$ consists of the attention projections (q_proj, k_proj, v_proj, o_proj) and MLP layers (gate_proj, down_proj, up_proj).
> * **Impact of G:** The number of groups $G$ is determined by the model architecture and is not treated as a tunable hyperparameter.
>
> However, the **Per-Rank KL** variant in our ablation study (Figure 2) can be viewed as implicitly increasing G by further subdividing each LoRA module into individual rank-level components. The results show that this finer-grained grouping performs slightly worse or comparable to PESO, suggesting that module-level grouping provides an robust  balance between stability and flexibility.
>
> # W3. Comparison with full parameter fine-tuning
> > The experimental results in Table 2 are missing a comparison to a standard full fine-tuning baseline.
>
> We have added a comparison between LoRA fine-tuning (Single Evolving LoRA) and full-parameter fine-tuning on the Instrument dataset. As shown in the table below, **LoRA is comparable to or outperforms full fine-tuning**.
> * **At high learning rates** ($2\mathrm{e}{-5}$): Full fine-tuning suffers from severe overfitting and catastrophic forgetting, leading to poor performance.
> * **At low learning rates** ($2\mathrm{e}{-6}$): Full fine-tuning improves but remains slightly worse or comparable to LoRA.
>
> This suggests that LoRA acts as a beneficial structural regularizer in continual settings, naturally mitigating forgetting by restricting the update space, whereas full fine-tuning is prone to over-adapting to the most recent block.
>
> | Method                               |  hit@5 | hit@10 | ndcg@5 | ndcg@10 |
> |--------------------------------------|-------:|-------:|-------:|--------:|
> | Full fine-tuning (lr = 2e-5)      | 0.0142 | 0.0228 | 0.0099 | 0.0127  |
> | Full fine-tuning (lr = 2e-6)     | 0.0171 | **0.0254** | 0.0122 | 0.0149  |
> | LoRA fine-tuning (lr = 2e-5)  | **0.0181** | 0.0253 | **0.0127** | **0.0150**  |
>
> ## Reference
> * [1] Wang et al., Structure Aware Incremental Learning with Personalized Imitation Weights for Recommender Systems, AAAI 2023
> * [2] Yoo et al., Embracing Plasticity: Balancing Stability and Plasticity in Continual Recommender Systems, SIGIR 2025
> * [3] Wang et al., Learnable Item Tokenization for Generative Recommendation, CIKM'24

---

> ### Author Response · Authors · 2025-12-01
> **Summary of Rebuttal**
>
> We thank Reviewer 4gyJ again for their positive assessment and constructive feedback. As there was no post-rebuttal discussion, we summarize below the key revisions and new analyses included in our updated paper to address their concerns:
>
> * **(W1) Quantifying Distribution Shift:** To validate the realism of our data splits, we explicitly quantified preference drift using a **domain discrimination metric**. The results show a steady increase in cumulative drift ($\delta(0,t)$ rises from 0.20 to 0.46), confirming that our setup reflects **realistic, accumulating structural shifts** rather than random noise.
> * **(W2) Clarification on Groups ($G$):** We clarified that groups correspond to **standard model modules** (e.g., attention projections). We linked our "Per-Rank KL" ablation to the granularity of $G$, demonstrating that module-level grouping provides a more robust balance between stability and flexibility than finer-grained partitions.
> * **(W3) Comparison with Full Fine-Tuning:** We added a **Full-Parameter Fine-Tuning** baseline. Experiments confirm that **LoRA is comparable to or outperforms Full Fine-Tuning** (Hit@5: **0.0181** vs. 0.0171). We found that Full Fine-Tuning suffers from a dilemma between catastrophic forgetting (at high LR) and insufficient adaptation (at low LR), whereas LoRA acts as an effective structural regularizer.
>
> **We believe these quantitative analyses and additional baselines fully address the reviewer's questions regarding the realism of the setting and the efficacy of LoRA. We hope these updates confirm the validity of our approach.**
>
> Best regards,
> The Authors

---

### Official Review · Reviewer_PErP · 2025-11-02

**Soundness:** 2
**Presentation:** 2
**Contribution:** 2
**Rating:** 2
**Confidence:** 4

**Summary:**

This paper studies continual learning in LLM-based generative recommender systems. The authors argue that existing LoRA-based continual learning methods emphasize preserving past performance but overlook that recommendation requires modeling evolving user preferences rather than retaining outdated ones. To address this, they propose PESO, which maintains a single evolving LoRA adapter regularized toward its previous state. Theoretically, the authors show that the proximal term provides data-aware, direction-wise guidance in the LoRA subspace. Empirical results on Amazon datasets demonstrate consistent improvement over existing LoRA-based continual learning methods.

**Strengths:**

1. The paper tackles continual learning in recommendation, a fundamental and practical problem.

2. The motivation is clear, with comprehensive analysis of single vs. cumulative LoRA and their limitations in the recommendation setting.

3. The theoretical justification for the proximal regularization providing data-aware, direction-wise guidance in the LoRA subspace is well presented.

4. Experiments are systematic and demonstrate consistent gains over baselines.

**Weaknesses:**

1. The motivation for using LoRA as the primary PEFT technique is not fully convincing. Comparison or discussion with alternatives such as prompt tuning or layer pruning is missing.

2. The paper lacks discussion and comparison with recent works on continual or incremental learning for LLM-based generative recommendation [1].

3. The experimental datasets lack diversity, all drawn from the Amazon Review corpus in the e-commerce domain, limiting the generalization of conclusions.

4. While the theoretical formulation is elegant, the intuitive explanation of how direction-wise guidance relates to long-term versus short-term preference adaptation could be elaborated.

[1] Shi et al., Preliminary Study on Incremental Learning for Large Language Model-based Recommender Systems. CIKM'24.

**Questions:**

1. What are the advantages of using LoRA compared to other parameter-efficient fine-tuning methods, such as prompt tuning?

2. Can the authors elaborate on the intuition of how data-aware, direction-wise guidance in the LoRA subspace helps balance long-term and evolving user preferences?

3. How would the method perform in non-e-commerce or multi-domain continual recommendation scenarios?

4. Is there any case studies on how the proposed method balances the long-term interests and current evolved user preferences?

---

> ### Author Response · Authors · 2025-11-21
> **Response to Reviewer PErP (Part 1)**
>
> **Dear Reviewer PErP,**
>
> **We sincerely thank you for your valuable feedback and for recognizing the strengths of our work! We have carefully revised the paper based on your suggestions and incorporated additional complementary experiments and analyses. Please refer to the updated version, where all revisions and additions are highlighted in blue for your convenience.**
>
> **Below we provide the point‑by‑point responses to all concerns (weaknesses and questions) raised in your review.**
>
> # Q1/W1. Advantages of LoRA over prompt tuning
> > What are the advantages of using LoRA compared to other parameter-efficient fine-tuning methods, such as prompt tuning?
>
> Prompt-tuning–based PEFT methods typically learn a prompt pool and dynamically retrieve the most relevant prompts for each input, inserting them into the input or intermediate representations without updating backbone weights. This introduces **inference overhead** because the model must compute query features and perform similarity matching over a growing prompt pool at inference time.
>
> The **inference-inefficiency** is even more severe in our generative recommendation setting: autoregressive generation requires many forward passes per prediction, and each step would need repeated prompt retrieval. Recent studies in vision [1,2] also report that LoRA-based methods generally outperform prompt-based approaches in large-scale tasks, making LoRA the preferred PEFT technique.
>
> We also implemented a simple L2P-style approach [3] that retrieves prompts and prepends them to intermediate layers, **but it performed worse than LoRA-based fine-tuning**:
>
> | Method             | hit@5  | hit@10 | ndcg@5 | ndcg@10 |
> |--------------------|--------|--------|--------|---------|
> | L2P prompt tuning  | 0.0154 | 0.0208 | 0.0107 | 0.0124 |
> | **LoRA fine-tuning**   | **0.0181** | **0.0253** | **0.0127** | **0.0150** |
>
>
> # W2. Comparison with CIKM'24 [4]
> > The paper lacks discussion and comparison with recent works on continual or incremental learning for LLM-based generative recommendation.
>
> Thank you for pointing out this relevant method. Following the reviewer's suggestion, we have compared CIKM'24 [4], its parameter-inheritance variant, and our PESO on the Instrument dataset. The CIKM’24 approach interpolates two LoRAs: one trained on historical data $D_0$ (e.g., the first 60% of whole data) and another trained only on new data ($D_1,...,D_4$). Our results show that **CIKM'24 consistently underperforms our PESO**, showing that simply combining historical and new LoRAs has limitations similar to cumulative LoRAs—being overly stable and insufficiently adaptive in recommendation settings. **Adding parameter inheritance offers a small gain, but it still falls short of PESO.** This highlights PESO’s more effective balance between stability and plasticity.
>
>
> | Method                             | hit@5  | hit@10 | ndcg@5 | ndcg@10 |
> |------------------------------------|-------:|-------:|-------:|--------:|
> | CIKM'24 [4]                            | 0.0164 | 0.0250 | 0.0117 | 0.0144  |
> | CIKM'24 with parameter inheritance | 0.0183 | 0.0254 | 0.0130 | 0.0153  |
> | **PESO**                               | **0.0193** | **0.0268** | **0.0138** | **0.0162** |
>
> # W3/Q3. Non e-commerce datasets
> > The experimental datasets lack diversity, all drawn from the Amazon Review corpus in the e-commerce domain, limiting the generalization.
>
>
> Following your suggestion to explore non-e-commerce domains, we evaluated PESO on the **Yelp dataset**, where interactions correspond to user check-ins at locations, using the same data-splitting strategy as in our main experiments. As shown in the table below, **PESO consistently outperforms strong competitors**, including Single Evolving LoRA and SumLoRA.
>
> This result is particularly notable given that, unlike Amazon products which feature detailed textual descriptions, Yelp locations often **lack deep semantic content** (consisting primarily of names like "Pizza Hut" or coarse categories like "Pizza" or "Restaurant"). This demonstrates the robustness of our method, showing it remains highly effective even in settings with limited semantic richness.
>
>
> | Methods               | hit@5  | hit@10 | ndcg@5 | ndcg@10 |
> |-----------------------|--------|--------|--------|---------|
> | Pretrain              | 0.0201 | 0.0309 | 0.0126 | 0.0161 |
> | Single Evolving LoRA  | 0.0290 | 0.0442 | 0.0190 | 0.0239 |
> | SumLoRA               | 0.0287 | 0.0432 | 0.0191 | 0.0238 |
> | **PESO**              | **0.0302** | **0.0454** | **0.0199** | **0.0248** |

---

> ### Author Response · Authors · 2025-11-21
> **Response to Reviewer PErP (Part 2)**
>
> # W4. Intuitive explanation of direction-wise guidance
> > While the theoretical formulation is elegant, the intuitive explanation of how direction-wise guidance relates to long-term versus short-term preference adaptation could be elaborated.
>
>
>
> We have revised the paper to provide a more intuitive explanation of how PESO provides **data-aware, direction-wise guidance**. Mathematically, each direction $q_k$ is an eigenvector of the gradient covariance $\Sigma_t$, representing a specific **pattern of parameter change**. Semantically, this acts as a **latent axis of user preference** (e.g., a "Sci-Fi affinity" or "Vintage style"), effectively isolating specific behaviors that the model can adjust independently.
>
> The eigenvalue $\sigma_k^2$ **determines the balance between adapting to the new optimum and preserving the previous state** along these axes:
>
> *  **Short-term Adaptation (Large $\sigma_k^2$):** If the data strongly supports a specific preference axis (e.g., a sudden increase in mystery books), PESO moves the parameters toward the new optimum $v_t^*$ along this direction, ensuring rapid plasticity.
> *  **Long-term Preservation (Small $\sigma_k^2$):** If the evidence for an axis is weak (e.g., a long-term interest in acoustic guitars is currently unobserved), PESO anchors the parameters to the previous state $v_{t-1}$, ensuring stability.
>
>
>
>
>
> # Q4. Case studies on how PESO balances stability and plasticity
> > Is there any case studies on how the proposed method balances the long-term interests and current evolved user preferences?
>
>
> To examine this balance, we analyzed the **final model's performance** (after training on all blocks) on the Instruments dataset, focusing on **two specific user groups** that serve as proxies for the conflicting goals of Long-term Stability and Short-term Plasticity:
> 1. **Dormant Users (The "Stability" Test)**: Users active in the past who return after a long gap ($D_0 \to D_4$). Success requires retaining **long-term interests** despite new updates.
> 2. **New Users (The "Plasticity" Test)**: Users appearing only in the final stage ($D_4$). Success requires rapidly adapting to **current evolved preferences** without prior history.
>
> **Case Study Results**: The table below clearly illustrates the trade-off:
> * **Single Evolving LoRA (High Plasticity)** excels at New Users but fails on Dormant Users due to catastrophic forgetting.
> * **Cumulative LoRA (High Stability)** (i.e., SumLoRA) retains Long-term interests but is too rigid to adapt to New Users.
> * **PESO (Effective Balance)** is the **only method that succeeds in both scenarios**. It achieves the highest performance on both groups, demonstrating that it can dynamically allocate plasticity to new signals while using proximal regularization to preserve long-term knowledge.
>
>
> | Methods     \ User groups     | New users | Dormant users |
> |----------------------------------|-----------|------------|
> | Single Evolving LoRA (Plasticity-focused) | 0.0116    | 0.0154     |
> | Cumulative LoRA (Stability-focused)               | 0.0101    | 0.0164     |
> | **PESO (Balanced)**  | **0.0122**| **0.0170** |
>
>
>
>
>
>
> ## Reference
> * [1] Liang and Li, InfLoRA: Interference-Free Low-Rank Adaptation for Continual Learning, ICLR 2024
> * [2] Wu et al., SD-LoRA: Scalable Decoupled Low-Rank Adaptation for Class Incremental Learning, ICLR 2025
> * [3] Wang et al., Learning to Prompt for Continual Learning, CVPR 2022
> * [4] Shi et al., Preliminary Study on Incremental Learning for Large Language Model-based Recommender Systems, CIKM 2024.

---

> ### Author Response · Authors · 2025-12-01
> **Summary of Rebuttal**
>
> We thank Reviewer PErP again for their valuable review. As there was no post-rebuttal discussion, we summarize below the key revisions and new experiments included in our updated paper to address their concerns:
>
> * **(Q1/W1) LoRA vs. Prompt Tuning:** We clarified that LoRA is preferred **over prompt tuning** for generative recommendation due to the **high inference latency** of retrieving prompts during autoregressive decoding. We also empirically saw thatPESO significantly outperforms  **L2P-style prompt tuning baseline**  (Hit@5: **0.0181** vs. 0.0154).
> * **(W2) Comparison with more baselines:** We implemented the requested **CIKM'24 baseline** (and its parameter-inheritance variant). Experiments on the Instruments dataset confirm that **PESO consistently outperforms both variants**, proving superior plasticity-stability balance.
> * **(W3/Q3) Generalization to Non-E-commerce dataset:** We added experiments on the **Yelp dataset** (location check-ins). Despite Yelp's limited textual richness compared to Amazon, **PESO maintained superior performance** over strong baselines, demonstrating robustness across domains.
> * **(W4, Q4) Intuition & Case studies:** We provided a more intuitive explanation to explicitly map eigenvalue magnitude to **"new" vs. "dormant"** interests. We validated this with a **User Group Analysis**, showing PESO is the **only method** to succeed on both "New Users" (Plasticity test) and "Dormant Users" (Stability test).
>
> **We believe these additional experiments and revisions fully address the concerns regarding baselines, non-e-commerce datasets, theoretical intuition, and case studies. We hope these updates warrant a re-evaluation of our work.**
>
> Best regards, The Authors

---

### Meta-Review · Area_Chair_xYZK · 2026-01-13

**Summary:**

This paper proposes PESO, a continual learning framework for LLM-based recommendation that addresses the stability–plasticity dilemma by using a single evolving LoRA with proximal regularization. Most reviewers acknowledged that the method is innovative and theoretically grounded. The main remaining discussions focused on whether the baseline comparisons were sufficient and whether the datasets used for model selection and validation were adequate. During the rebuttal phase, the authors also provided additional experiments and results to address these concerns.

**Reviewer Concerns:**

1. Insufficient baseline comparisons. The method needs to be compared against state-of-the-art continual learning algorithms for recommendation, as well as additional training strategies, to better demonstrate its superiority.

2. Lack of generalization validation. The current experimental datasets consist of different categories within the same scenario, which does not sufficiently demonstrate the effectiveness or generalizability of the proposed approach.

3. Missing methodological details. Some key components of the method are not described clearly in the paper and require further clarification.

**Reviewer Scores:**

During the rebuttal phase, the authors provided more comprehensive experimental analyses, including comparisons with the SOTA baselines highlighted by the reviewers and additional training methods. They also introduced multiple new datasets to validate the effectiveness and generalization of the approach, obtaining consistently positive results. Furthermore, several methodological details were clarified. Overall, I believe these responses are likely to be well received by the reviewers.

---

### Decision · Program_Chairs · 2026-01-26

Accept (Poster)